# Prevalence of questionable research practices, research misconduct and their potential explanatory factors: A survey among academic researchers in The Netherlands

Gowri Gopalakrishna[1]*, Gerben ter Riet[2], Gerko Vink[3‡], Ineke Stoop[4‡], Jelte M. Wicherts[5], Lex M. Bouter[1,6]

1 Department of Epidemiology and Data Science, Amsterdam University Medical Centers, Amsterdam, The Netherlands, 2 Faculty of Health, Center of Expertise Urban Vitality Amsterdam University of Applied Science, Amsterdam, The Netherlands, 3 Department of Methodology & Statistics, Utrecht University, Utrecht, The Netherlands, 4 The Netherlands Institute for Social Research, Den Haag, The Netherlands, 5 Department of Methodology and Statistics, Tilburg University, Tilburg, The Netherlands, 6 Department of Philosophy, Faculty of Humanities, Vrije Universiteit Amsterdam, Amsterdam, The Netherlands

☯ These authors contributed equally to this work.
‡ GV and IS also contributed equally to this work.
* g.gopalakrishna@amsterdamumc.nl

**Data Availability Statement:** All data files are available from the Open Science Platform (https://osf.io/dp6zf/). DOI: 10.17605/OSF.IO/DP6ZF.

## Abstract

Prevalence of research misconduct, questionable research practices (QRPs) and their associations with a range of explanatory factors has not been studied sufficiently among academic researchers. The National Survey on Research Integrity targeted all disciplinary fields and academic ranks in the Netherlands. It included questions about engagement in fabrication, falsification and 11 QRPs over the previous three years, and 12 explanatory factor scales. We ensured strict identity protection and used the randomized response method for questions on research misconduct. 6,813 respondents completed the survey. Prevalence of fabrication was 4.3% (95% CI: 2.9, 5.7) and of falsification 4.2% (95% CI: 2.8, 5.6). Prevalence of QRPs ranged from 0.6% (95% CI: 0.5, 0.9) to 17.5% (95% CI: 16.4, 18.7) with 51.3% (95% CI: 50.1, 52.5) of respondents engaging frequently in at least one QRP. Being a PhD candidate or junior researcher increased the odds of frequently engaging in at least one QRP, as did being male. Scientific norm subscription (odds ratio (OR) 0.79; 95% CI: 0.63, 1.00) and perceived likelihood of detection by reviewers (OR 0.62, 95% CI: 0.44, 0.88) were associated with engaging in less research misconduct. Publication pressure was associated with more often engaging in one or more QRPs frequently (OR 1.22, 95% CI: 1.14, 1.30). We found higher prevalence of misconduct than earlier surveys. Our results suggest that greater emphasis on scientific norm subscription, strengthening reviewers in their role as gatekeepers of research quality and curbing the "publish or perish" incentive system promotes research integrity.

**Funding:** - Awarded to Lex M. Bouter Grant No.: 20-22600-98-401 Netherlands Organisation for Health Research and Development (ZonMw) https://www.zonmw.nl/en/news-and-funding/news/detail/item/largest-study-ever-on-research-integrity-launches-aimed-at-all-researchers-in-the-netherlands/ The funders had no role in study design, data collection and analysis, decision to publish, or preparation of the manuscript. - Awarded to Jelte M. Wicherts Grant No.: Consolidator Grant 726361 (IMPROVE) European Research Council (ERC) https://erc.europa.eu he funders had no role in study design, data collection and analysis, decision to publish, or preparation of the manuscript.

**Competing interests:** The authors have declared that no competing interests exist.

## Introduction

The basis of sound public policy relies on trustworthy and high quality research [1]. This trust is earned by being transparent and by performing research that is relevant, replicable, ethically sound and of rigorous methodological quality. Yet trust in research and replicability of previous findings [2] are compromised by researchers engaging in research misconduct, such as fabrication and falsification (FF) and subtle trespasses of ethical and methodological principles [3]. Continued efforts to promote responsible research practices (RRPs) which include open science practices like open data sharing, pre-registration of study protocols, open access publication over questionable research practices (QRPs) are therefore needed. In order to support the need for such continued efforts, solid evidence on the prevalence of research misconduct and QRPs as well as the factors promoting or curtailing such behaviours are needed.

QRPs include subtle trespasses such as not submitting valid negative results for publication, not reporting flaws in study design or execution, selective citation to enhance one's own findings and so forth. The global discussion of the 'replication crisis' [2] has highlighted common worries about these QRPs becoming alarmingly prevalent and suggests underlying systematic factors, such as increased publication and funding pressures and lowered behavioural norms. After several major cases of misconduct [4], the global research community is converging to a common view on ways to foster research integrity [5].

While many integrity promoting initiatives exist [3, 6–8], strong evidence on which factors prevent these trespasses is lacking. The studies addressing this [9–13] are discipline-specific and focus on few factors to explain the occurrence of QRPs and FF. A broad range of explanatory factors such as scientific norm subscription, organizational justice in terms of distribution of resources and promotions, competition, work, publication and funding pressures, and mentoring need to be considered in order to comprehensively understand the occurrence of QRP incidence [14–17]. The National Survey on Research Integrity (NSRI) [18] targets the prevalence of QRPs, FF and RRPs as well as their postulated explanatory factors. It targets all academic researchers in The Netherlands across all disciplinary fields and uses a randomized response (RR) technique to assess engagement in FF as it is a well-validated method known to elicit more honest answers on highly sensitive topics [19].

NSRI's objectives are to estimate:

1. disciplinary field-specific prevalence of QRPs, FF and RRPs;

2. associations between explanatory factors and QRPs, FF and RRPs

In this paper, we focus on the NSRI results on QRPs, FF and postulated explanatory factors. Elsewhere [20], we report on our findings on RRPs and their postulated explanatory factors.

## Materials and methods

### Ethics approval

The Ethics Review Board of the School of Social and Behavioral Sciences of Tilburg University approved the NSRI (Approval Number: RP274). The Dutch Medical Research Involving Human Subjects Act was deemed not applicable by the Institutional Review Board of the Amsterdam University Medical Centers (Reference Number: 2020.286).The full NSRI questionnaire, its raw anonymized dataset, the complete data analysis plan, its source codes and version controls of the analysis (displayed in Github) can be found on the Open Science Framework [21].

### Study design

The NSRI is a cross-sectional study using a web-based anonymized questionnaire. All academic researchers working at or affiliated to at least one of 15 universities or 7 University

Medical Centers in The Netherlands were invited by email to participate. To be eligible, researchers had, on average, to do at least 8 hours of research-related activities weekly <u>and</u> belong to life and medical sciences;social and behavioural sciences; natural and engineering sciences; or the arts and humanities; <u>and</u> be a PhD candidate or junior researcher (who is defined in The Netherlands as an individual with a Masters or PhD degree doing a minimum of 8 hours per week of research related tasks under close supervision)postdoctoral researcher or assistant professor; or associate or full professor.

The survey was conducted by a trusted third party, Kantar Public [22] which is an international market research company that adheres to the ICC/ESOMAR International Code of Standards [23]. Kantar Public's sole responsibility was to send the survey invitations and reminders by email to our target group and, at the end of the data collection period, send the research team the anonymized dataset.

Universities and University Medical Centers that supported NSRI supplied Kantar Public with the email addresses of their eligible researchers. Email addresses for the other institutes were obtained through publicly available sources, such as university websites and PubMed.

Researchers' informed consent was sought through a first email invitation which contained the survey link, an explanation of NSRI's purpose and its identity protection measures. Consenting invitees could immediately participate. NSRI was open for data collection for seven weeks, during which three reminder emails were sent to non-responders, at a one to two week interval period. Only after the full data analysis plan had been finalized and preregistered on the Open Science Framework [21], Kantar Public sent us the anonymized dataset containing individual responses.

## Survey instrument

NSRI comprises of four components: 11 QRPs, 11 RRPs, two FFs and 12 explanatory factor scales (75 questions, detailed in S6 Table). The survey started with a number of background questions to assess eligibility of respondents. These included questions on one's weekly average duration of research-related work, one's dominant field of research, academic rank, gender and if one was doing empirical research or not [21].

All respondents obtained the same set of questions on QRPs, RRPs and FF, referring to one's behavior in the previous three years. A three year timeframe was chosen to limit recall bias and is also a timeframe used in other similar studies [9, 10]. The 11 QRPs were adapted from a recent study where 60% of the surveyed participants came from the biomedical disciplinary field [24]. As the NSRI targeted disciplinary fields including those outside of the biomedical field, we conducted a series disciplinary field specific focus groups to ensure the 11 QRPs from Bouter et al. were applicable to our multidisciplinary target group. All QRPs had 7-point Likert scales ranging from 1 to 7 where 1 = never and 7 = always (no intermediate linguistic labels were used) plus a "not applicable" (NA) answer option. The two FF questions used the RR technique with only a yes or no answer option [25]. The RR technique is best known to elicit more honest answers, the more sensitive in nature the questions are [19, 25]. Additionally, because the technique takes longer to apply, the survey would end up taking too long when all questions would use the technique. Hence, we chose to limit its use to only the most sensitive questions on research misconduct.

The explanatory factors scales were based on psychometrically tested scales most commonly used in the research integrity literature and focused on action-ability. Twelve were selected: scientific norms, peer norms, perceived work pressure, publication pressure, pressure due to dependence on funding, mentoring (responsible and survival), competitiveness of the research field, organizational justice (distributional and procedural), and likelihood of QRP detection by collaborators and reviewers [16, 24, 26–30]. Some of the scales were incorporated into the

NSRI questionnaire verbatim, others were adapted for our population or newly created (see S5 Table). The scales on scientific norms, peer norms, competitiveness, organizational justice (procedural and distributional), and perceived likelihood of QRP detection by collaborators and reviewers were piloted. The other exploratory factor scales were either used previously in highly similar samples (e.g. publication pressure scale) [27] or in samples in earlier studies which were sufficiently similar to our current sample [31, 32] except for the funding pressure scale which was newly created but could not be piloted due to resource constraints. However, in the NSRI, this scale performed well in terms of psychometric properties (with a Cronbach's alpha of 0.76) and in terms of convergent validity (i.e., positive correlations with publication pressure and competitiveness [S4 Table]).

We used "missingness by design" to minimize survey completion time. Thus, each invitee received one of three random subsets of 50 explanatory factor items from the full set of 75 (see S5 Table). All explanatory factor items had 7-point Likert scales. In addition, the two perceived likelihood of QRP detection scales, the procedural organizational justice scale and the funding pressure scale had a "not applicable" (NA) answer option. There was no item non-response as respondents had to either complete the survey or withdraw. We pre-tested the NSRI questionnaire's comprehensibility in cognitive interviews [15] with 8 academics from different ranks and disciplines. In summary, the comments centered around improvement in layout such as the removal of an instruction video on the RR technique which was said to be redundant, improvement in the clarity of the instructions and to emphasize certain words in the questionnaire by use of different fonts for improved clarity. The full report of the cognitive interview can be accessed at the Open Science Framework [21].

## Statistical analysis

In this paper, we focus on three outcomes: (i) overall mean QRP, (ii) prevalence of any frequent QRP and (iii) any FF. The associations of these three outcomes with the five background characteristics (S1 Table) and the explanatory factor scales (Table 1) were investigated with multiple (i) linear regression, (ii) binary logistic regression and (iii) ordinal logistic regression, respectively [17]. Mean scores of individual QRPs only consider respondents that deemed the QRP at issue applicable meaning for each of the QRP columns, mean scores were calculated only over values 1–7 and "not applicable" answers were not part of this calculation. In the multiple linear regression analysis (Tables 3 and 4), overall mean QRP was computed as the average score on the 11 QRPs, after recoding not applicable scores to 1 (i.e. never). Prevalence was operationalized as the percentage of respondents who scored at least one QRP as 5, 6 or 7 among the respondents for that QRP. This definition allows for comparability to other studies [9, 10]. S2A–S2E Fig show the distribution of responses for the 11 QRPs. The label 'any FF' was assigned if a respondent had admitted to at least one instance of falsification or fabrication.

For the multivariable analyses of the explanatory factor scales we used z-scores computed as the first principal component of the corresponding items [14]. Missing explanatory factor item scores due to 'not applicable' answers were replaced by the mean z-score of the other items of the same scale. Multiple imputation with mice in R (version 4.0.3) was employed to deal with the missingness by design [33, 34]. Fifty complete data sets were generated by imputing the missing values using predictive mean matching [35, 36]. The regression models were fit to each of the 50 datasets, and the results combined into a single inference. To incorporate uncertainty due to the nonresponse, the standard errors were computed according to Rubin's Rules [37]. All multivariable models contain the five background variables and the explanatory factor scales. The subscales distributional and procedural organizational justice were highly correlated (correlation factor of >0.8 [S4 Table]). They were thus merged to gain precision leading

**Table 1. Mean scores (standard deviations) and z scores of explanatory factor scales stratified by disciplinary field and academic rank.**

| Explanatory factor scale | Disciplinary field | | | | Academic rank | | | Overall |
|---|---|---|---|---|---|---|---|---|
| | Life and medical sciences | Social and behavioural sciences | Natural and engineering sciences | Arts and humanities | PhD candidates and junior researchers | Postdocs and assistant professors | Associate and full professors | |
| **Work pressure** | 4.5 (1.3) | 4.5 (1.4) | 4.4 (1.4) | 4.8 (1.4) | 3.9 (1.3) | 4.7 (1.3) | 4.8 (1.4) | 4.5 (1.4) |
| *Cronbach's alpha: 0.79* | 0.00 | 0.01 | -0.10 | 0.20 | -0.43 | 0.16 | 0.21 | 0.00 |
| **Publication pressure** | 3.8 (1.2) | 4.0 (1.2) | 3.9 (1.2) | 4.1 (1.3) | 3.8 (1.2) | 4.2 (1.2) | 3.7 (1.2) | 3.9 (1.2) |
| *Cronbach's alpha: 0.80* | -0.06 | 0.05 | 0.00 | 0.13 | -0.07 | 0.21 | -0.21 | 0.00 |
| **Funding pressure** | 4.8 (1.4) | 4.6 (1.4) | 4.8 (1.4) | 4.6 (1.4) | 4.1 (1.5) | 5.2 (1.2) | 4.7 (1.3) | 4.7 (1.4) |
| *Cronbach's alpha: 0.76* | 0.05 | -0.13 | 0.05 | -0.10 | -0.38 | 0.28 | -0.06 | -0.01 |
| **Mentoring (survival)** | 4.0 (1.4) | 3.8 (1.5) | 3.8 (1.5) | 3.6 (1.5) | 4.1 (1.4) | 4.00 (1.5) | 3.5 (1.5) | 3.9 (1.5) |
| *Cronbach's alpha: 0.90* | 0.08 | -0.02 | -0.06 | -0.17 | 0.13 | 0.06 | -0.21 | 0.00 |
| **Mentoring (responsible)** | 4.1 (1.5) | 3.9 (1.5) | 3.8 (1.5) | 3.4 (1.6) | 4.4 (1.4) | 3.8 (1.5) | 3.5 (1.5) | 3.9 (1.5) |
| *Cronbach's alpha: 0.91* | 0.13 | -0.03 | -0.07 | -0.34 | 0.34 | -0.03 | -0.28 | 0.00 |
| **Competitiveness** | 3.6 (1.0) | 3.5 (1.0) | 3.5 (1.0) | 3.8 (1.0) | 3.4 (0.9) | 3.7 (1.0) | 3.6 (1.0) | 3.6 (1.0) |
| *Cronbach's alpha: 0.70* | 0.03 | -0.06 | -0.06 | 0.19 | -0.20 | 0.10 | 0.06 | 0.00 |
| **Scientific norms** | 6.1 (0.6) | 6.1 (0.6) | 6.2 (0.6) | 6.2 (0.6) | 5.9 (0.7) | 6.2 (0.6) | 6.2 (0.6) | 6.1 (0.6) |
| *Cronbach's alpha: 0.71* | -0.04 | -0.01 | 0.07 | 0.06 | -0.29 | 0.07 | 0.19 | 0.00 |
| **Peer norms** | 4.2 (0.9) | 4.2 (0.9) | 4.4 (1.0) | 4.1 (1.0) | 4.3 (1.0) | 4.1 (0.9) | 4.3 (1.0) | 4.2 (1.0) |
| *Cronbach's alpha: 0.84* | -0.03 | -0.06 | 0.17 | -0.09 | 0.05 | -0.12 | 0.11 | 0.00 |
| **Organizational justice** ** | 4.4 (1.1) | 4.3 (1.2) | 4.5 (1.2) | 3.9 (1.3) | 4.6 (1.0) | 4.1 (1.1) | 4.5 (1.2) | 4.4 (1.2) |
| *Cronbach's alpha: 0.91* | 0.02 | -0.01 | 0.14 | -0.31 | 0.14 | -0.20 | 0.15 | 0.00 |
| **Likelihood of detection (collaborators)** | 3.6 (1.0) | 3.5 (1.0) | 3.6 (1.0) | 3.5 (1.1) | 3.5 (1.1) | 3.5 (1.0) | 3.7 (1.0) | 3.6 (1.0) |
| *Cronbach's alpha: 0.65* | 0.03 | -0.05 | 0.04 | -0.06 | -0.04 | -0.05 | 0.11 | 0.00 |
| **Likelihood of detection (reviewers)** | 4.2 (1.2) | 4.3 (1.2) | 4.4 (1.2) | 4.2 (1.3) | 4.1 (1.3) | 4.3 (1.2) | 4.4 (1.1) | 4.3 (1.2) |
| *Cronbach's alpha: 0.83* | -0.07 | 0.04 | 0.05 | -0.06 | -0.14 | 0.00 | 0.08 | 0.00 |

Scales ranging from 1 (never, totally disagree, very unlikely) to 7 (always, totally agree, very likely)

**Two subscales (distributional and procedural organizational justice) were merged due to high correlation; S4 & S5 Tables show the correlation of all the explanatory factor scales and scale items.

to the formation of one Organizational Justice scale. Results in S4 Table demonstrate that the correlations for the separate subscales were highly similar to those obtained from combining these scales. The full statistical analysis plan, and statistical analysis codes were preregistered on the Open Science Framework [21].

## Identity protection

Respondents' identity protection was ensured in accordance to the European General Data Protection Regulation (GDPR) and corresponding legislation in The Netherlands as follows:

first, Kantar Public conducted the survey to ensure that the email addresses of respondents were never handled by the research team. Second, Kantar Public did not store respondents' URLs and IP addresses. The anonymized dataset was sent to the research team upon closure of data collection and preregistration of the statistical analysis plan. Third, we used the RR method for the two most sensitive questions [25]. RR creates a probabilistic and not a direct association between a respondent's answer and the pertinent behaviour, adding an additional layer of confidentiality. Finally, we conducted analyses at aggregate levels only, that is across disciplinary fields, gender, academic rank, whether respondents conducted empirical research and were employed by an NSRI-supporting research institution (see S1 Table).

## Results

### Descriptive analyses

Of the 22 universities and University Medical Centers in the Netherlands, eight supported the NSRI. A total of 63,778 emails were sent out (Fig 1) of which 9529 eligible respondents started the survey after passing the screening questions and 6813 completed it. The percentage response could only be reliably calculated for the supporting institutions (S1A Fig). This is 21.2%. S1 Table describes these respondents, stratified by background characteristics.

There are about equal proportions of male and female respondents. Further breakdown by disciplinary field, academic rank, research type and institutional support is detailed in S1 Table. Of respondents in the natural and engineering sciences, 24.9% are women. In the rank of associate and full professors, women make up less than 30% of respondents (S1 Table). Nearly 90% of all respondents are engaged in empirical research. Respondents from support- ing and non-supporting institutions are fairly evenly distributed across disciplinary fields and academic ranks, except for the natural and engineering sciences where less than one in four

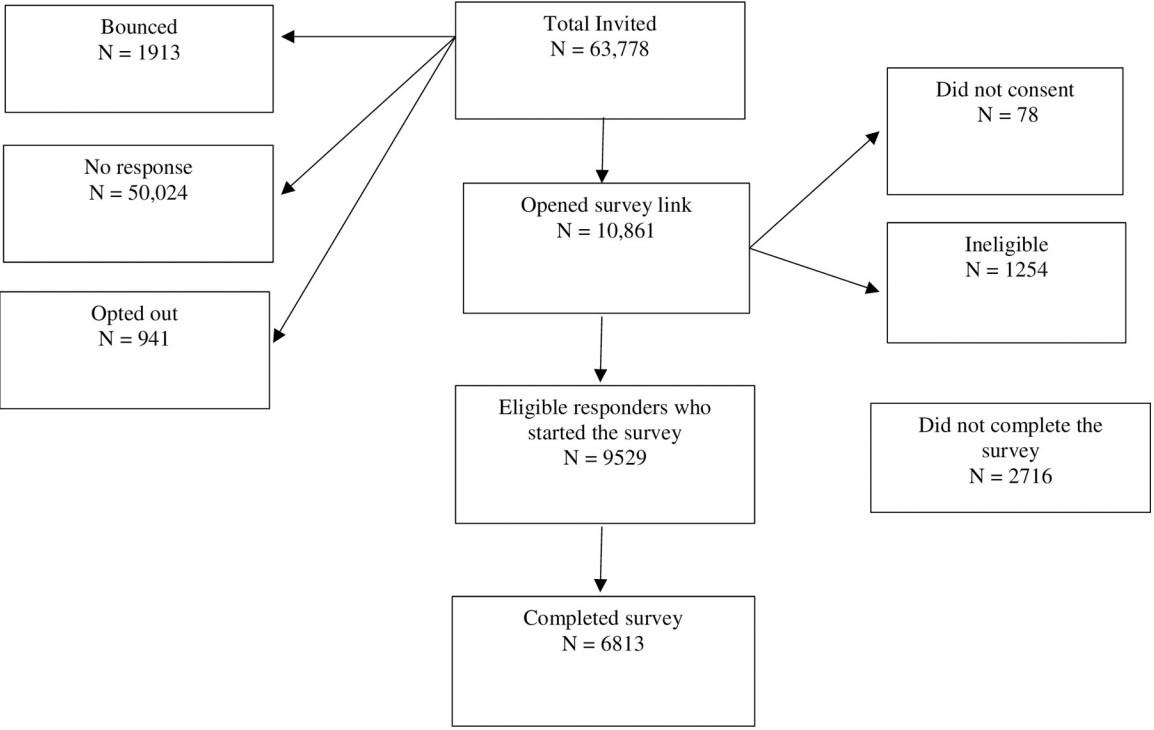

**Fig 1. Flow chart of the survey.**

(23.5%) come from supporting institutions. Postdocs and assistant professors report the highest scale scores for publication pressure (4.2), funding pressure (5.2) and competitiveness (3.7), and the lowest scale score for peer norms (4.1) and organizational justice (4.1) when compared to the other academic ranks (Table 1). Respondents from the arts and humanities have the highest scale scores for work pressure (4.8), publication pressure (4.1) and competitiveness (3.8). They also have the lowest scores for mentoring, peer norms organizational justice (3.5, 4.1 and 3.9, respectively) when compared to the other disciplinary fields (Table 1). The scientific norms scale scores, although much higher than the peer norms scale scores, show a similar trend of higher scientific norm scores and lower peer norm scores, across disciplinary fields and academic ranks.

## Prevalence of QRPs and research misconduct

Table 2 shows the prevalence of the QRPs and FFs. The five most prevalent QRPs (i.e. Likert scale score 5, 6 or 7) are: (i) "Not submitting or resubmitting valid negative studies for publication" (QRP 9: 17.5%), (ii)"Insufficient inclusion of study flaws and limitations in publications" (QRP 10: 17%), (iii) "insufficient supervision or mentoring of junior co-workers" (QRP 2: 15%), (iv) "insufficient attention to the equipment, skills or expertise" (QRP 1: 14.7%), and (v) "inadequate note taking of the research process" (QRP 7: 14.5%) (Table 2, Fig 2). Less than 1% of respondents said they unfairly reviewed manuscripts, grant applications or colleagues (QRP 4: 0.8%) or engaged in "improper referencing of sources" frequently (QRP 6: 0.6%) in the last three years.

"Not (re)submitting valid negative studies for publication" (QRP 9) has the highest prevalence of "not applicable" (NA) across all disciplines with the arts and humanities on top (72.3%) (S2 Table). About one in two PhD candidates and junior researchers (48.7%) reported QRP 4 (i.e. "unfairly reviewed manuscripts, grant applications or colleagues") as not applicable to them. Overall, the arts and humanities scholars have the highest prevalence of NAs for nine out of the 11 QRPs. PhD candidates and junior researchers have the highest NA prevalence for 10 out of 11 QRPs (S2 Table). This group also has the highest prevalence for 8 out of 11 QRPs across ranks (Table 2).

Respondents from the life and medical sciences have the highest prevalence of any frequent QRP compared to the other disciplinary fields (55.3%, Table 2). The life and medical sciences respondents also have the highest prevalence estimate for any FF (10.4%). Less than 1% of arts and humanities scholars reported fabrication. However, for falsification, these scholars have the highest prevalence estimate (6.1% 95% CI: 1.4, 10.9; Table 2).

## Regression analyses

Tables 3 and 4 show the results of the regression analyses for the five background characteristics and the explanatory factor scales, respectively. All models include the five background characteristics and all explanatory factor scales.

Table 3 shows that being a PhD candidate or a junior researcher is associated with a statistically significantly higher odds of any frequent QRP. Being non-male (i.e. female or gender undisclosed) and doing non-empirical research is associated with a lower overall QRP mean and lower odds of any frequent QRP. The associations of the background characteristics with any FF have wide 95% confidence intervals and none are statistically significant.

Table 4 shows that a standard deviation increase on the publication pressure scale is associated with an increase of 0.10 in the overall QRP mean score. Similarly, each standard deviation increase on the scientific norms, peer norms and organizational justice scales is associated with a lower overall QRP mean scores of 0.12, 0.04, and 0.04, respectively (Table 4).

**Table 2. Prevalence (95% confidence intervals) of the QRPs, any frequent QRP and fabrication or falsification stratified by disciplinary field and academic rank.**

| QRP | Description (In the last three years.) | Disciplinary field | | | | | Academic rank | | | |
| --- | --- | --- | --- | --- | --- | --- | --- | --- | --- | --- |
| | | Life and medical sciences | Social and behavioural sciences | Natural and engineering sciences | Arts and humanities | PhD candidates and junior researchers | Postdocs and assistant professors | Associate and full professors | Overall |
| QRP1 | Insufficient attention to the equipment, skills or expertise | 15.2 (13.9,16.7) | 14.7 (13.0,16.5) | 13.4 (11.6,15.4) | 16.2 (13.0,20.0) | 15.9 (14.2,17.7) | 14.6 (13.2,16.1) | 13.7 (12.2,15.4) | 14.7 (13.8,15.7) |
| QRP2 | Insufficiently supervised or mentored junior co-workers | 16.1 (14.7,17.6) | 13.8 (12.1,15.6) | 14.9 (13.0,17.0) | 13.4 (10.5,16.9) | 12.9 (11.1,14.9) | 14.4 (13.0,15.8) | 17.0 (15.4,18.7) | 15.0 (14.1,15.9) |
| QRP3 | Inadequate research designs or unsuitable measurement instruments | 4.4 (3.7,5.3) | 4.6 (3.7,5.7) | 4.3 (3.3,5.6) | 2.9 (1.6,5.0) | 6.0 (4.9,7.2) | 4.0 (3.3,4.9) | 3.2 (2.5,4.1) | 4.3 (3.9,4.9) |
| QRP4 | Unfairly reviewed manuscripts, grant applications or colleagues | 0.7 (0.4,1.2) | 0.9 (0.5,1.5) | 1.1 (0.6,1.9) | 0.4 (0.1,1.6) | 1.2 (0.6,2.1) | 0.6 (0.3,1.0) | 0.9 (0.5,1.4) | 0.8 (0.6,1.1) |
| QRP5 | Conclusions not sufficiently substantiated | 3.7 (3.0,4.5) | 4.0 (3.2,5.1) | 4.3 (3.3,5.5) | 4.9 (3.3,7.1) | 6.1 (5.0,7.3) | 3.5 (2.9,4.3) | 2.8 (2.2,3.7) | 4.0 (3.6,4.5) |
| QRP6 | Improper referencing of source | 0.6 (0.4,1.0) | 0.4 (0.2,0.8) | 0.9 (0.5,1.6) | 0.8 (0.3,2.0) | 1.1 (0.7,1.7) | 0.6 (0.3,1.0) | 0.3 (0.1,0.7) | 0.6 (0.5,0.9) |
| QRP7 | Inadequate notes of research process | 13.8 (12.5,15.2) | 14.4 (12.8,16.2) | 16.1 (14.1,18.3) | 14.6 (11.5,18.3) | 15.0 (13.4,16.7) | 15.0 (13.7,16.5) | 13.4 (11.8,15.1) | 14.5 (13.7,15.5) |
| QRP8 | Failed to report important study details in publications | 2.9 (2.3,3.7) | 3.0 (2.3,3.9) | 2.4 (1.7,3.4) | 2.9 (1.7,5.0) | 3.1 (2.3,4.0) | 2.6 (2.1,3.4) | 2.9 (2.2,3.8) | 2.8 (2.4,3.3) |
| QRP9 | Not submitting or resubmit valid negative studies for publication | 14.5 (13,16.2) | 17.2 (15.1,19.5) | 25.3 (22.3,28.5) | 19.9 (14.4,26.7) | 17.1 (14.8,19.6) | 19.5 (17.6,21.4) | 15.5 (13.7,17.5) | 17.5 (16.4,18.7) |
| QRP10 | Insufficient inclusion of study flaws and limitations in publications | 17.8 (16.4,19.4) | 17.2 (15.5,19.1) | 15.8 (13.9,17.9) | 15.2 (12.1,19) | 21.2 (19.3,23.3) | 16.9 (15.5,18.4) | 13.7 (12.2,15.3) | 17.0 (16.1,18.0) |
| QRP11 | Selectively cited references to enhance findings or convictions | 15.8 (14.5,17.3) | 11.8 (10.4,13.4) | 13.8 (12.1,15.8) | 13.4 (10.9,16.5) | 20.0 (18.2,22) | 13.5 (12.2,14.9) | 9.5 (8.3,10.9) | 14.0 (13.2,14.9) |
| Any frequent QRP | Score 5, 6 or 7 on at least 1 of the 11 QRPs | 55.3 (53.4, 57.1) | 50.2 (48.0, 52.5) | 49.4 (46.8, 52.0) | 42.1 (38.3, 46.1) | 52.5 (50.3, 54.7) | 52.3 (50.4, 54.2) | 48.9 (46.7, 51.0) | 51.3 (50.1, 52.5) |
| Fabrication | Making up of data or results | 5.5 (3.2, 7.7) | 4.8 (2.2, 7.5) | 2.5 (0, 5.5) | 0.7 (0, 5.1) | 4.0 (1.4, 6.6) | 4.9 (2.6, 7.1) | 3.6 (1.1, 6.1) | 4.3 (2.9, 5.7) |
| Falsification | Manipulating research materials, data or results | 4.9 (2.7, 7.2) | 2.0 (0, 4.6) | 5.3 (2.2, 8.4) | 6.1 (1.4, 10.9) | 5.5 (2.8, 8.1) | 2.6 (0.4, 4.8) | 5.3 (2.7, 7.9) | 4.2 (2.8, 5.6) |
| Any FF | Fabrication and/or Falsification | 10.4 (7.1, 13.7) | 5.7 (1.8, 9.5) | 7.6 (3.1, 12.1) | 8.4 (1.6, 15.3) | 8.9 (5.0, 12.6) | 7.3 (4.1, 10.6) | 8.9 (5.1, 12.7) | 8.3 (6.2, 10.3) |

Prevalence is based on the QRP at issue having a Likert score of 5, 6 or 7 among respondents that deemed the QRP at issue applicable; Any frequent QRP is based on the presence of at least one of the 11 QRPs; All figures in this table are percentages and refer to the last 3 years.

**Table 3. Regression coefficients and odds ratios (95% confidence interval) of overall QRP mean, any frequent QRP and any FF stratified by five background characteristics.**

| | | Overall QRP Mean[^] | Any Frequent QRP[¶] | Any FF[#] |
|---|---|---|---|---|
| | | Linear regression model[††] coefficient (95% CI) | Logistic regression model[††] OR (95% CI) | Ordinal regression model[††] OR (95% CI) |
| **Disciplinary field** *Reference category: Life and medical sciences* | Social and behavourial sciences | -0.09 (-0.13, -0.05) | 0.81 (0.72, 0.92) | 0.82 (0.44, 1.50) |
| | Natural and engineering sciences | -0.07 (-0.11, -0.03) | 0.80 (0.69, 0.92) | 0.92 (0.47, 1.79) |
| | Arts and humanities | -0.25 (-0.31, -0.19) | 0.61 (0.50, 0.74) | 1.18 (0.53, 2.63) |
| **Academic rank** *Reference category: Postdocs and assistant professors* | PhD candidates and junior researchers | 0.03 (-0.01, 0.07) | **1.16 (1.01, 1.32)** | 0.94 (0.49, 1.79) |
| | Associate and full professors | -0.01 (-0.05, 0.02) | 0.95 (0.84, 1.08) | 1.54 (0.82, 2.86) |
| **Gender** *Reference category: Male* | Female | **-0.09 (-0.12, -0.06)** | **0.77 (0.69, 0.85)** | 1.26 (0.73, 2.16) |
| | Undisclosed | **-0.18 (-0.29, -0.07)** | **0.65 (0.45, 0.96)** | 1.00 (0.30, 3.27) |
| **Engaged in empirical research** *Reference category: Yes* | No | **-0.15 (-0.20, -0.10)** | **0.76 (0.64, 0.91)** | 0.63 (0.28, 1.46) |
| **Institutional support** *Reference category: No* | Yes | -0.03 (-0.06, 0.00) | 0.93 (0.83, 1.03) | 1.09 (0.64, 1.85) |

[^] Overall mean QRP was computed as the average score on the 11 QRPs with the not applicable scores recoded to 1 (i.e. never)

[¶] Any frequent QRP is defined as at least one of the 11 QRPs having a score of 5, 6 or 7 on the Likert scale

[#] Any FF refers to fabrication or falsification

[††] All models contain the five background characteristics (see Table 3) and all 10 explanatory factor scales; Bold figures are statistically significant.

**Table 4. Regression coefficients and odds ratios (95% confidence interval) of overall QRP mean, any frequent QRP and any FF stratified by explanatory factor scales.**

| | Overall QRP mean[^] | Any frequent QRP[¶] | Any FF[#] |
|---|---|---|---|
| | Linear regression model[††] coefficient (95% CI) | Logistic regression model[††] OR (95% CI) | Ordinal regression model[††] OR (95% CI) |
| **Work pressure** | **0.02 (0.00, 0.04)** | 1.04 (0.98, 1.11) | 0.93 (0.67, 1.31) |
| **Publication pressure** | **0.10 (0.08, 0.12)** | **1.22 (1.14, 1.30)** | 1.09 (0.75, 1.59) |
| **Funding pressure** | 0.01 (-0.01, 0.03) | 1.01 (0.94, 1.08) | 1.06 (0.74, 1.54) |
| **Mentoring (survival)** | **0.04 (0.02, 0.06)** | 1.00 (0.93, 1.07) | 0.97 (0.66, 1.43) |
| **Mentoring (responsible)** | **-0.02 (-0.04, 0.00)** | 1.01 (0.94, 1.09) | 1.06 (0.71, 1.59) |
| **Competitiveness** | **0.02 (0.00, 0.04)** | 1.04 (0.98, 1.12) | 1.13 (0.79, 1.62) |
| **Scientific norm** | **-0.12 (-0.13, -0.10)** | **0.88 (0.83, 0.93)** | **0.79 (0.63, 1.00)** |
| **Peer norms** | **-0.03 (-0.05, -0.02)** | **0.91 (0.86, 0.97)** | 1.20 (0.85, 1.65) |
| **Organizational justice** ** | **-0.04 (-0.06, -0.02)** | **0.91 (0.85, 0.98)** | 0.96 (0.67, 1.38) |
| **Likelihood of detection (collaborators)** | 0.01 (-0.01, 0.03) | 0.99 (0.93, 1.06) | 0.96 (0.63, 1.48) |
| **Likelihood of detection (reviewers)** | 0.00 (-0.02, 0.02) | 0.99 (0.93, 1.06) | **0.63 (0.44, 0.88)** |

[^] Overall mean QRP was computed as the average score on the 11 QRPs with the not applicable scores recoded to 1 (i.e. never)

[¶] Any frequent QRP is defined as at least one of the 11 QRPs having a score of 5, 6 or 7 on the Likert scale

[#] Any FF refers to fabrication or falsification

[††] All models contain the five background characteristics (see Table 3) and all 10 explanatory factor scales

** Two subscales (distributional and procedural organizational justice) were merged due to high correlation; Bold figures are statistically significant.

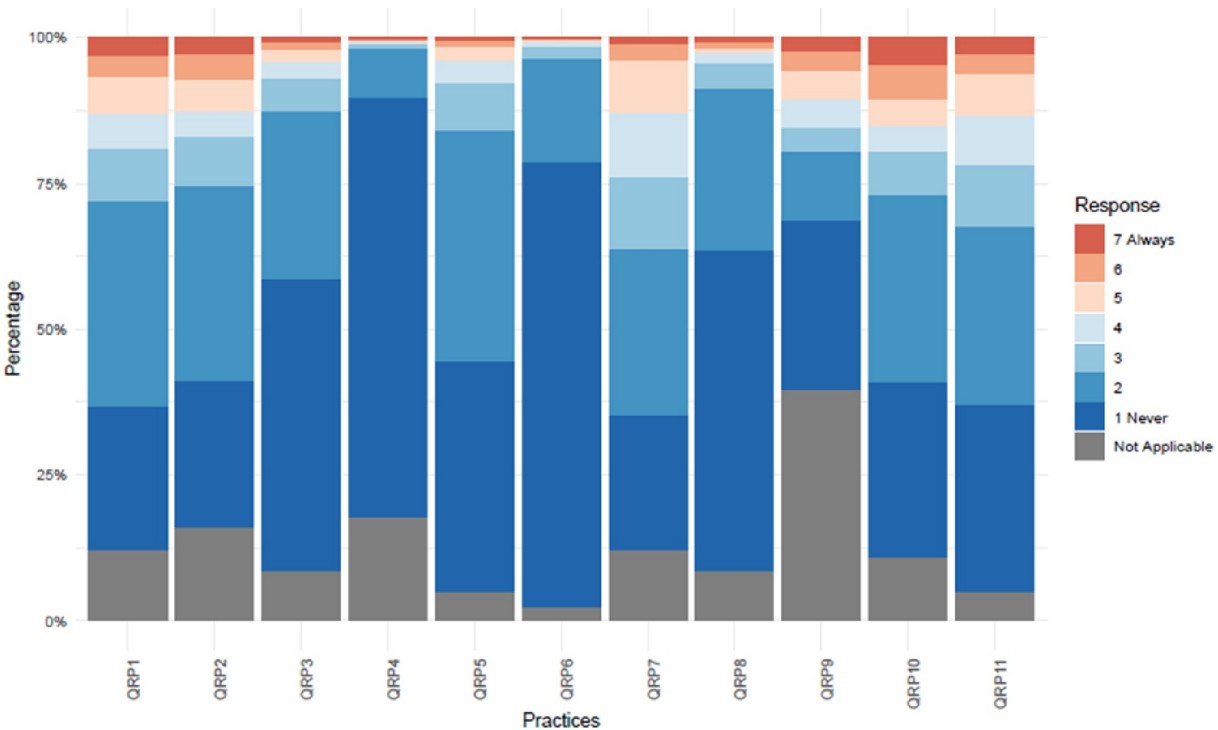

**Fig 2. Percentage of observed answer categories of QRPs across 6813 respondents.**

Logistic regression shows that for each standard deviation increase on the publication pressure scale, the odds of any frequent QRP increases by a factor of 1.22, while scientific norms subscription, peer norms and organizational justice scales worked the other way around for these three explanatory factors, i.e. the odds of any frequent QRP decreases by a factor of 0.88 (scientific norms), 0.91 (peer norms) and 0.91 (organizational justice), respectively.

Ordinal regression shows that for each standard deviation increase on scientific norms subscription or perceived likelihood of detection by reviewers scale, the odds of any FF decreases by a factor 0.79 and 0.62, respectively (Table 4).

## Discussion

### Summary of main findings

Our research integrity survey among academics across all disciplinary fields and ranks is one of the largest worldwide [9, 10]. Here, we share our findings on QRPs, fabrication and falsification as well as the explanatory factor scales that may be associated with the occurrence of these research misbehaviours. We find that over the last three years one in two researchers engaged frequently in at least one QRP, while one in twelve reported having falsified or fabricated their research at least once.

Postdocs and assistant professors rate publication pressure, funding pressure and competitiveness higher than other academic ranks, but peer norms and organizational justice lower. Arts and humanities scholars reported experiencing the highest work and publication pressures, the most competition and the lowest in mentoring, peer norms and organizational justice compared to other disciplinary fields. PhD candidates and junior researchers engage more often in any frequent QRP than other academic ranks as do males and those doing empirical as opposed to those doing non-empirical research.

Scientific norm subscription was the explanatory factor scale associated with the lowest prevalence of any frequent QRP and any FF. We also found that higher perceived likelihood of QRP detection by reviewers was associated with less FF. More publication pressure was associated with higher odds of any frequent QRP. Surprisingly, work pressure and competitiveness were only marginally associated with higher QRP mean while mentoring was only weakly negatively associated with overall mean QRP and not at all with the odds of any frequent QRP or any FF.

## Explanatory factors that may drive or reduce research misbehaviour and misconduct

Publication pressure appears to lead to the largest increase in the odds of any frequent QRP. This finding supports recent initiatives to change the "publish or perish" reward system in academia [26, 27, 38].

Our findings on the discrepancy between subscription to scientific norms espoused by respondents and their perceived adherence to such norms by their peers corroborate earlier findings in a study among 3600 researchers in the USA [15, 16]. Previous researchers have made calls to institutional leaders and department heads to pay increased attention to these scientific norms in order to improve adherence and promote responsible conduct of research [16, 28]. Scientific norms subscription was one of two explanatory factor scales with the largest significant association in lowering any frequent QRP and FF in our regression analyses.

Perceived likelihood of detection by reviewers is significantly associated with lower odds of any FF suggesting that reviewers may have an important role in preventing research misconduct. The increased transparency offered by open science practices such as data sharing, is likely to boost chances of detection of research misconduct whether through formal journal reviewers or otherwise such as through post publication peer review or other types of scholarly reviews such as comments on preprints [31].

Lack of proper supervision and mentoring of junior co-workers was one of the three most prevalent QRPs. A recent study of 1080 researchers in Amsterdam reported similar findings [32]. Unsurprisingly, we find a moderate yet statistically significant association between survival mentoring and higher overall QRP mean suggesting that survival mentoring may be associated with higher QRPs while an association in the opposite direction, again moderate but significant, is observed for responsible mentoring and lower overall QRP mean. Both results as expected. and reported in an earlier study [13] which explored five different types of mentoring (including responsible and survival mentoring that we measured). Our study and that of Anderson et al. [13] suggests that mentors can influence behaviour in ways that both increase (in the case of survival mentoring) or decrease (in the case of responsible mentoring) the likelihood of problematic research behaviours such as QRPs.

## Areas of focus within disciplines, academic ranks and gender

Lower perceived organizational justice among the arts and humanities has been previously reported [32]. This disciplinary field also has the highest proportion of NAs for nine out of the 11 QRPs, suggesting that what is deemed as a QRP in the selection of 11 we have chosen for the NSRI may differ within the arts and humanities.

Among academic ranks, we find that being a PhD candidate or junior researcher is associated with the a higher odds of engaging in any frequent QRP. This rank also has the highest prevalence for eight out of the 11 QRPs we measured. A recent Dutch study of academics postulated that this may be in part explained by the consistent lack of good supervision and mentoring of junior researchers [32]. The authors suggest that it is plausible that young researchers

may be more prone to unintentionally committing QRP given their lack of research experience in combination with poor supervision.

Additionally, a research environment where mistakes cannot be openly discussed may further deter newcomers from admitting errors made. A safe and supportive learning environment with adequate supervision is increasingly recognized as key in this regard [38]. The need to focus on PhD candidates or junior researchers is again emphasized as these researchers reported 10 of the 11 QRPs as being not applicable. While some QRPs are indeed rank-specific such as QRPs 2 and 4 on supervision and review of grant proposals respectively, the remaining nine are not rank-specific. Our finding that identifying as male is associated with higher odds of any frequent QRP and higher overall mean QRP agrees with findings by others [39, 40].

## QRP and FF prevalence

The prevalence of any frequent QRP was 51.3% which suggests that QRP may be more prevalent than previously reported. In other research integrity surveys, prevalence of self-reported QRPs were in the range of 13–33% [9, 10]. Our finding of a high prevalence of any frequent QRP might be due to the cut-off we used in our analysis, that is at least one QRP with a score of 5, 6 or 7 (with 1 being never and 7 being always). As other studies have used different cut-offs, answer scales and different number of QRPs and QRP definitions it render results between such surveys as not directly incomparable [9, 10]. However, a recent systematic review of surveys on research integrity showed that papers published after 2011 reported higher prevalence of misbehavior [9] which may be due to the increased awareness of research integrity in recent years although this cannot be ascertained conclusively.

When it comes to misconduct, previous surveys report the prevalence to be in the range of about 2–3% [9, 10] rising to as much as 15.5% when the questions concern misconduct observed in others [9]. In our study, the prevalence estimate of self–reported fabrication is 4.3% and self-reported falsification, 4.2%, while the prevalence estimate of any FF is 8.3%. When looking at disciplinary field-specific estimates of misconduct, life and medical sciences have the highest estimate of any FF (10.4%). These numbers are concerning and only comparable to one other smaller study (n = 140) that also used the RR technique [41]. This study found that 4.5% of their respondents admitted falsification. They did not assess fabrication [41].

The higher prevalence estimate of any FF in the life and medical sciences has been previously reported by others [10]. Unfortunately, it cannot be concluded if this is due to more misconduct actually taking place or because researchers in this particular disciplinary field are simply more aware of the issue and thus more willing to report it.

## Strengths and limitations

The email addresses of researchers affiliated to non-NSRI-supporting institutions were web-scraped from open sources. Therefore, we are unable to credibly verify if the scraped email addresses matched our eligibility criteria prior to participation in the survey. Hence, we could only reliably calculate the response to the NSRI based on the eight supporting institutions. The 21.1% response is within the range of similar research integrity surveys [10, 32]. Given this response, one may wonder how representative the NSRI sample is of the target population i.e. all academic researchers in the Netherlands. Unfortunately, there are no reliable numbers at the national level that match our study's eligibility criteria. Therefore, we cannot assess our sample's representativeness even for the five background characteristics. Nevertheless, we believe our results to be valid as our main findings align well with the findings of other research integrity surveys [13, 16, 28, 31, 32]. Furthermore, prevalence estimates of fabrication

and falsification may be more valid than those reported previously [9, 10] due to the use of the RR technique [19].

A limitation of our analysis concerns recoding NA answers into "never" for the multiple linear regressions since there is a difference between not committing a behaviour because it is truly not applicable and intentionally refraining from doing so. Our analyses may therefore underestimate the occurrence of true intentional QRPs. We have studied other recodes of the NA answers and remain confident that our preregistered choice yields inferences that do not ignore the non-random distributions of the NA answers and do not violate theoretical and practical expectations about the relation between QRP and other studied practices. Another limitation is our definition of "any frequent QRP", which we assigned to scores of 5, 6 or 7 on the Likert scale. Widening the definition of 'frequent' would have resulted in higher prevalence estimates. Furthermore, other surveys assessed a different number of QRPs and defined them sometimes differently, hampering direct comparisons between our survey and others.

Another potential limitation we wish to mention is misclassifications in academic rank due to promotion of individuals to a higher rank less than 3 years prior to completing our survey. Their responses are therefore likely to partly represent their behaviors whilst in a lower academic rank. However we did not collect information on years of experience of respondents in a rank due to strict privacy design of the survey. As such we are unable to comment on the impact of this misclassification on our results but we believe it to be relatively minor. Future surveys on this topic may, however, wish to take this into account in their design and analysis. The NSRI is the largest research integrity survey in academia to-date that has looked at not only prevalence of QRPs and FF but also at the largest range of possible explanatory factors in one single study across all disciplinary fields and academic ranks using the RR technique [19].

As a follow up to the NSRI, we plan to conduct in-depth interactive workshops to further understand the major drivers or suppressors of QRPs and FF in order to elucidate the nuances that a survey cannot capture.

## Supporting information

**S1 Fig.** a. Flowchart of supporting institutions (n = 8). b. Flowchart of non-supporting institutions (n = 14).
(DOCX)

**S2 Fig.** a. Percentage of observed answer categories of QRPs stratified by disciplinary field. b. Percentage of observed answer categories of QRPs stratified by academic rank. c. Percentage of observed answer categories of QRPs stratified by gender. d. Percentage of observed answer categories of QRPs stratified by research type (empirical Y/N). e. Percentage of observed answer categories QRPs stratified by institutional support (Y/N).
(TIF)

**S3 Fig. Scatter plot of mean explanatory factor scale scores by disciplinary field and academic rank.**
(TIF)

**S1 Table. Characteristics of all respondents by disciplinary field, academic rank, gender, research type and institutional support.**
(DOCX)

**S2 Table. Prevalence (%) of the "not applicable" answers stratified by disciplinary field and academic rank.**
(DOCX)

**S3 Table a Mean score (95% confidence interval) of QRPs stratified by disciplinary field and academic rank b Mean score (95% confidence interval) and prevalence (95% confidence interval) of QRPs stratified by gender, research type and institutional support.**
(DOCX)

**S4 Table. Correlation matrix of the z scores of the principal component analysis of the explanatory factor scales.**
(DOCX)

**S5 Table. Full list of the explanatory factor scales and their corresponding items showing which were adapted, newly created or piloted.**
(DOCX)

**S6 Table. National Survey on Research Integrity questionnaire.**
(DOCX)

## Acknowledgments

The authors wish to thank the NSRI Steering Committee members (Guy Widdershoven, Herman Paul, Joeri Tijdink, Sonja Zuijdgeest, Corrette Ploem) for their support. In addition, we wish to thank Sara Behrad, Frank Gerritse, Coosje Veldkamp, Brian Martinson and Melissa Anderson for their contributions.

## Author Contributions

**Conceptualization:** Gowri Gopalakrishna, Gerben ter Riet, Ineke Stoop, Jelte M. Wicherts, Lex M. Bouter.

**Data curation:** Gowri Gopalakrishna, Gerben ter Riet, Gerko Vink.

**Formal analysis:** Gowri Gopalakrishna, Gerben ter Riet, Gerko Vink, Ineke Stoop, Jelte M. Wicherts, Lex M. Bouter.

**Funding acquisition:** Gerben ter Riet, Jelte M. Wicherts, Lex M. Bouter.

**Investigation:** Gowri Gopalakrishna, Gerben ter Riet, Gerko Vink, Ineke Stoop, Jelte M. Wicherts, Lex M. Bouter.

**Methodology:** Gowri Gopalakrishna, Gerben ter Riet, Gerko Vink, Ineke Stoop, Jelte M. Wicherts, Lex M. Bouter.

**Project administration:** Gowri Gopalakrishna.

**Resources:** Gowri Gopalakrishna, Lex M. Bouter.

**Software:** Gerko Vink.

**Supervision:** Gowri Gopalakrishna, Gerben ter Riet, Jelte M. Wicherts, Lex M. Bouter.

**Validation:** Gowri Gopalakrishna, Gerben ter Riet, Gerko Vink, Jelte M. Wicherts, Lex M. Bouter.

**Visualization:** Gowri Gopalakrishna, Gerben ter Riet, Gerko Vink, Jelte M. Wicherts, Lex M. Bouter.

**Writing – original draft:** Gowri Gopalakrishna.

**Writing – review & editing:** Gowri Gopalakrishna, Gerben ter Riet, Gerko Vink, Ineke Stoop, Jelte M. Wicherts, Lex M. Bouter.

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
