## [Decision Letter · Decision Letter 0]

20 Oct 2021

PONE-D-21-28638Prevalence of questionable research practices, research misconduct and their potential explanatory factors: a survey among academic researchers in The NetherlandsPLOS ONE

Dear Dr. Gopalakrishna,

Thank you for submitting your manuscript to PLOS ONE. After careful consideration, we feel that it has merit but does not fully meet PLOS ONE’s publication criteria as it currently stands. Therefore, we invite you to submit a revised version of the manuscript that addresses the points raised during the review process.

Your manuscript was reviewed by three accomplished scientists. I also reviewed the paper and I concur with the vast majority of all comments, critiques and suggestions raised by the reviewers. This is a highly important manuscript that presents much needed findings in an area seldom studied.

We look forward to receiving your revised manuscript.

Kind regards,

Sergi Fàbregues

Academic Editor

PLOS ONE

Journal Requirements:

Reviewers' comments:

Reviewer's Responses to Questions

**Comments to the Author**

1. Is the manuscript technically sound, and do the data support the conclusions?

Reviewer #1: Yes

Reviewer #2: Yes

Reviewer #3: Partly

2. Has the statistical analysis been performed appropriately and rigorously? 

Reviewer #1: Yes

Reviewer #2: Yes

Reviewer #3: Yes

3. Have the authors made all data underlying the findings in their manuscript fully available?

Reviewer #1: Yes

Reviewer #2: Yes

Reviewer #3: Yes

4. Is the manuscript presented in an intelligible fashion and written in standard English?

Reviewer #1: Yes

Reviewer #2: Yes

Reviewer #3: Yes

5. Review Comments to the Author

Reviewer #1: The paper provides more than 6,000 responses to a survey to understand QRPs and research misconduct. It is a pre-registered study in the Open Science Framework and the materials that complement the article add value to it. Overall I find that both the introduction and discussion are well structured and highlight important literature and highlight their findings with previous work. The analyses chosen and the results seem adequate, however my main concern is that it is difficult to follow the thread of what has been done. I suggest that both the methods section and the presentation of results could be improved for publication of the article.

(1) It is convenient to add the psychometric properties of the explanatory factor scale scores (Not only Cronbach’s alpha) in this sample and not to rely only on previous studies. If this is not possible due to the length of the manuscript, perhaps it can be added as part of S6. Along the same lines, I also missed the results of the cognitive interviews (line 136).

(2) I recommend that in the section on statistical analysis: first present the three outcomes (as it is now) are presented. Then, it is confusing to me to say that there is a multiple linear regression that has never been presented. Perhaps it would be preferable to anticipate the analyses and then detail them. Perhaps the details of how the scores were obtained could be part of the "survey instrument" section.

(3) The work seen in the osf is not reflected in the description of the analysis. In addition to improving the presentation of the analyses, I think it would be good to complement them with more information.

(4) Please, justify in methods section why distributional and procedural organizational justice have been merged. It seems a very serious issue for only mention it in a table footer.

Below you will also find other minor details.

Supplementary material is not ordered in presentation order and sometimes is confusing.

Line 51. I think that in this case “QRP” should be QRPs.

Line 68. RRP has been defined already in line 50.

Line 70. In line 1, randomized response (RR) is mentioned but its importance is not explained. Later it is discussed again and here it is explained what it is. I would suggest eliminating RR here or anticipating its importance.

Line 89 to 93. I suggest adding a semicolon to separate the field of specialization from the position held.

Line 117. Why is it necessary to adapt de QRPs?

Line 134. Please define NA (like in 152). Not all readers are familiar with this nomenclature.

Line 144. Why QRP has been dichotomized? Are there precedents of other authors doing it? If so, please give references.

150. What multivariate analysis are you referring to?

154 Number 50 seems a reference. Maybe write “Fifty”

Line 182. I would add “see S1 table” for better reading.

S1 table. Give more space in order to see full words in columns. Considering adding the N inside brackets: Life and medical sciences (N = 2746). It seems that in the cells there is the N (Column) and also de % (row).

S1a. In title add spaces in n = 8

S1b. In al flowchart add space after N = NUMBER.

S1c and S2a to s2e. The image is pixelated. Maybe raising the dpi can make it better.

S2b. Add sample size of subgroups. Probably this helps understanding the CI95% of undisclosed.

Table 1. “Chronbach’s alpha” is “Cronbach’s alpha”. Conisdering reducing the space between columns, and maybe add acronyms for better visualization.

Table 2. Please always use 1 decimal. For example, QRP1 in Arts and humanities 95%IC needs a 0.

Reviewer #2: In the manuscript titled ‘Prevalence of questionable research practices, research misconduct and their potential explanatory factors: A survey among academic researchers in The Netherlands’ (PONE-D-21-28638), the authors conducted a study using the National Survey on Research Integrity to examine the prevalence of research misconduct, questionable research practices, and how these are associated with a variety of explanatory factors across various disciplinary fields and academic ranks in the Netherlands. The article is well-written, and the supplementary materials are helpful in further examining the survey and analyses. This article has potential to contribute to the existing literature on questionable research practices, however there are some issues that should be addressed.

Introduction

The introduction is well-written, and authors have provided a sound rationale for why this research is important. The authors also provided detailed explanations about the NSRI and its objectives in assessing QRPs, FF, and RRP.

Materials and Methods

The authors provided sufficient detailed information on the ethical review board approval. One advantage of this study is that the authors used the Open Science Framework to share their study protocol, ethics approvals, data analysis plan, and final dataset. This was helpful and important for reviewing this study, and for readers to gain a more in-depth understanding of the study questions and analysis plan.

Study Design

The authors provided clear inclusion criteria of eligible participants for the study and the ways eligible participants were notified about the study.

Survey Instrument

The authors presented clear explanations of the NSRI, the different components, and types of response options for the different sets of questions. It would be helpful to briefly summarize the types of questions for each of the components on page 6 lines 115-120, similar to the example provided for the explanatory factor scales.

Another advantage of this study was that authors used the randomized response technique to reduce social desirability and nonresponse. It could be helpful to briefly explain why the RR technique was only used with the FF questions (page 6 lines 118-120).

Page 6 lines 122-126: Twelve explanatory variables including: scientific norms, peer norms, perceived work pressure, etc. were chosen. It would be beneficial to include reasons for why these twelve were chosen. Also, how many total explanatory variables are originally identified from the NSRI? It is not clear if the explanatory factors are comprised of 12 variables or if these were specifically chosen by the authors for a particular reason. Moreover, additional information would be important to include on the reason(s) why only the scientific norms description and normative behavior of peers were piloted (page 6, lines 128-129).

The authors conducted cognitive interviews to pre-test the NSRI comprehensibility, which can contribute to improving the survey. It would be interesting to note whether critical points were identified from the cognitive interviews that led to changes on the NSRI.

Statistical analysis

Information on item analysis (e.g., whether items were deleted, modified, replaced due to low item-total correlation or item discrimination) are not provided. In addition, Cronbach’s alpha values are provided to examine the reliability of the explanatory factor scale, but not with the other items/scales. This information would be beneficial to include across the different components of the NSRI or provide a rationale on why this was not included.

Minor comment: Page 7 lines 140-141 states that “Mean scores of individual QRPs only consider respondents that deemed the QRP at issue applicable.” It would be helpful to know the total number of participants who deemed the QRP an issue.

Results

Page 9 line 182 states: “There are about equal propositions of male and female respondents.” It would be beneficial to expand on this by including additional descriptives on the sample composition such as total number of participants across gender, disciplinary field, academic rank, and, if available, mean age, and average years spent in each academic rank.

Page 9 lines 188-190 states: “Postdocs and assistant professors report the highest scale score for publication pressure (4.2), funding pressure (5.2), and competitiveness (3.7), and the lowest scale score for organizational justice (4.1) (Table 1).” However, based on Table 1, it appears that the highest scale scores for postdocs and assistant professors are: scientific norms (6.2), funding pressure (5.2), and work pressure (4.7), and the lowest scale score for this group was likelihood of detection (collaborators) (3.5). In addition, it also appears that the highest scale scores for the respondents from the arts and humanities includes scientific norms (6.2), work pressure (4.8), and funding pressure (4.6), with the lowest including mentoring (3.5) and likelihood of detection (collaborators) (3.5).

Minor comment: It would be beneficial to include a description on how junior researcher was operationalized for the purposes of the study (e.g., two years post PhD).

Discussion

If results from page 9 lines 188-190 are revised, page 14 lines 280-281 should also be revised to reflect these changes (e.g., postdocs and assistant professors report highest scale score to be scientific norms, funding pressure, and work pressure).

Strengths and Limitations

Authors noted critical limitations to the study (e.g., coding NA responses as ‘never’ for statistical analyses and definition of “any frequent QRP” response associated with scores of 5, 6, or 7). These are important limitations that could potentially affect the results of the study.

Reviewer #3: Dear Authors,

Thank you for your interest in PLOS ONE. I very much enjoyed reading the manuscript as it emphasizes on the areas for improvement in research practices in academe. I have some suggestions and feedback for your consideration.

General comments:

• You have many abbreviations, which, I think, makes it hard to read. I would suggest no more than 2-3 abbreviations throughout the manuscript to help your reader follow your ideas.

• Line 154, you start a sentence with a number. Please consider writing it out.

Introduction:

• Line 47: The very first sentence is extremely broad and does not give us enough information about the kind of “important challenges.” Please explicate that part after challenges to prepare your reader about the topic.

• Line 50: There is a transition sentence needed after the sentence ends with “…methodological principals” and before “To promote responsible research practices…” to make a connection between two different ideas. In addition (Line 50), you start the sentence with “responsible research practices” and “questionable research practices” without any explanation about its meaning. I would suggest explaining these concepts before you move on to the problem and gap.

Study Design:

• I had hard time to understand your classification of participants as PhD candidate or junior researcher, postdoctoral researcher or assistant professor, or associate or full professor. A justification (along with a reference) is needed to explain how and why you considered this categorization and what you mean by “junior researcher.” Do you mean researchers, such as data analysts, who have degrees at master’s level OR you paid attention to their years of experience as researcher? Clarification is needed.

• Regarding your classification of participants based on their status (e.g., PhD candidate, postdoc, etc.), your survey involves items that asked about their actions over the previous three years, which needs justification (please consider adding a brief sentence for why the last three years not five, for example). So, when you report your findings, you actually report on their previous positions. For example, whatever a postdoc provided answers about in the survey actually refers to his/her activities as a PhD candidate unless the postdoc has the postdoc position for more than 3 years. Therefore, not only what their status (e.g., assistant professor, post doc) is but also the years they have been in that position matters and affects the interpretation of findings. Otherwise, whole findings are misleading. If you have their years of experience in that position, please consider adding that variable in the analysis to account for retrospective effect and report accordingly. If you do not have that variable in the survey, meaning no data collected about their years in their current position, then I would suggest acknowledging this as limitation and adding a note when interpreting findings.

Survey Instrument:

• Line 119, you note that randomized response technique was used but you do not explain what it is until later in the manuscript (Lines 375-376). I would suggest moving that explanation to the survey instrument section to help your reader understand your justification and what this technique is/does.

• Lines 135-137, you provide a statement about the pre-testing the NSRI questionnaire through cognitive interviews. This should be moved up on that page before you explain the number of total items and other details (starting after the sentence Line 129 or adding a paragraph before Line 130). I would suggest adding a short paragraph and explaining whether you made any changes on the questionnaire based on cognitive interviews. If that information was captured in your previous study, then consider citing it with a brief explanation.

Statistical Analysis:

• I had hard time to understand why you chose to compute average score on the 11 QRPs after recoding not applicable scores to 1. That skews your findings and does not accurately reflect participants’ attitudes because it was not applicable to them. I would suggest re-running the analysis after coding not applicable as zero (0). It should not be included in the sum or average score. You can explain this with a brief sentence and note that range of total score in the survey.

Results:

• Line 176: You mention eight universities supported the NSRI but do not provide information about non-supporting universities until later in the manuscript. An explanation about why they did not support, if known, would be helpful to contextually understand the findings as there might be some bias or buy-in to the work you did, which affects your findings as this cannot be generalizable in the context of the Netherlands.

• A brief description of the word “prevalence” would be helpful in the context of the study to help reader understand what you mean. Line 203, you note “The five most prevalent QRPs…” where you mean highly scoring items overall, whereas Line 57, you note about “QRPs becoming alarmingly prevalent” where you emphasize the commonness of the problem. These are two different things and need to be clarified.

Discussion:

• The first sentence (Lines 274-275) needs citations of others’ work.

• Line 310, “….journal reviewers or otherwise” what do you mean by “otherwise?”

• Lines 314-317, add a sentence after this sentence to explain the connection between the earlier study mentioned and your study.

• The concept of “sloppy science” is used with and without quotation mark. Is there a reference for this concept in using quotation mark?

• Line 356: the statement needs citation as you reference to a smaller study.

Strengths and limitations:

• Lines 365-366, you note that you were unable to credibly verify eligibility criteria based on emails. Didn’t you ask eligibility questions? I could not understand why this would be an issue if you asked eligibility questions.

• Lines 366-367, you note that you calculated responses only based on eight supporting institutions. Then, what is the point of reporting non-supporting institutions on Lines 185-188?

Wishing you all the best with next steps.

6. PLOS authors have the option to publish the peer review history of their article (what does this mean?). If published, this will include your full peer review and any attached files.

Reviewer #1: No

Reviewer #2: No

Reviewer #3: **Yes: **Sinem Toraman

---

## [Author Response · Author response to Decision Letter 0]

15 Nov 2021

15 Nov 2021

Authors’ responses to reviewers' Questions

Comments to the Author

1. Is the manuscript technically sound, and do the data support the conclusions?

Reviewer #1: Yes

Reviewer #2: Yes

Reviewer #3: Partly

2. Has the statistical analysis been performed appropriately and rigorously? 

Reviewer #1: Yes

Reviewer #2: Yes

Reviewer #3: Yes

3. Have the authors made all data underlying the findings in their manuscript fully available?

Reviewer #1: Yes

Reviewer #2: Yes

Reviewer #3: Yes

4. Is the manuscript presented in an intelligible fashion and written in standard English?

Reviewer #1: Yes

Reviewer #2: Yes

Reviewer #3: Yes

5. Review Comments to the Author

Reviewer #1: 

The paper provides more than 6,000 responses to a survey to understand QRPs and research misconduct. It is a pre-registered study in the Open Science Framework and the materials that complement the article add value to it. Overall I find that both the introduction and discussion are well structured and highlight important literature and highlight their findings with previous work. The analyses chosen and the results seem adequate, however my main concern is that it is difficult to follow the thread of what has been done. I suggest that both the methods section and the presentation of results could be improved for publication of the article.

(1) It is convenient to add the psychometric properties of the explanatory factor scale scores (Not only Cronbach’s alpha) in this sample and not to rely only on previous studies. If this is not possible due to the length of the manuscript, perhaps it can be added as part of S6.

Thank you for this comment. In response to this request we have now added the corrected item-total correlations for each of the explanatory factor scales in the renumbered S5 Table of the revised manuscript. 

Along the same lines, I also missed the results of the cognitive interviews (line 136).

Results on the cognitive interviews were centered around improving the clarity of certain words in the questionnaire, instructions, as well as feedback on the duration taken to answer the questionnaire. We have now added this information which reads as follows:

“In summary, the comments centered around improvement in layout such as the removal of an instruction video on the RR technique which was said to be redundant, improvement in the clarity of the instructions and to emphasize certain words in the questionnaire by use of different fonts for improved clarity. The full report of the cognitive interview can be accessed at the Open Science Framework [30]. (lines 162-166)

(2) I recommend that in the section on statistical analysis: first present the three outcomes (as it is now) are presented. Then, it is confusing to me to say that there is a multiple linear regression that has never been presented. Perhaps it would be preferable to anticipate the analyses and then detail them. 

Thank you for this comment and we apologize for the confusion. We wish to point out that the multiple linear regressions are presented in Tables 3a and 3b. We have made this more explicit now in lines 175-177 which now also make reference to Tables 3a and 3b from the Results section. 

“In the multiple linear regression analysis (Tables 3a and 3b), overall mean QRP was computed as the average score on the 11 QRPs, after recoding not applicable scores to 1 (i.e. never).”

Perhaps the details of how the scores were obtained could be part of the "survey instrument" section.

Thank you for this suggestion. To clarify, for the QRP columns, means scores for the QRPs were calculated only over Likert scale scores of 1-7 and “not applicables” were not part of this calculation.

All statistical analyses, code book and corresponding output are detailed in the OSF https://osf.io/ehx7q/ also referenced as reference 30 in the References section of this manuscript. They contain details on how the scores were computed. In the interest of word limit, we have only briefly described this in this section. We have modified lines 172-175 to read as follows:

“Mean scores of individual QRPs only consider respondents that deemed the QRP at issue applicable meaning for each of the QRP columns, mean scores were calculated only over values 1-7 and “not applicable” answers were not part of this calculation.”

(3) The work seen in the osf is not reflected in the description of the analysis. In addition to improving the presentation of the analyses, I think it would be good to complement them with more information.

Thank you for this suggestion. We have carefully checked the materials and methods section once again against the data analysis documents listed in the OSF. We can confirm that all aspects in the OSF are also mentioned in the subsection “Statistical analysis”. To further clarify all components in the OSF folder, we have added the following lines:

“The full NSRI questionnaire, its raw anonymized dataset, the complete data analysis plan, its source codes and version controls of the analysis (displayed in Github) can be found on the Open Science Framework [30].” (lines 90-92)

(4) Please, justify in methods section why distributional and procedural organizational justice have been merged. It seems a very serious issue for only mention it in a table footer.

Thank you for raising this point. Results in S4 Table demonstrate that the correlations for the separate subscales yielded correlations that were highly similar to those obtained from the combined scales. These subscales since highly correlated (S4 Table) and although merging them would be a deviation from our data analysis plan, we did so to gain precision. We have now included a clearer explanation of this in the Materials and Methods section in addition to the Table footers which reads as follows:

“The subscales responsible mentoring and survival mentoring as well as the subscales distributional and procedural organizational justice were highly correlated (correlation factor of >0.8 (S4 Table)). They were thus merged to gain precision leading to the formation of Mentoring and Organizational Justice scales respectively. Results in S4 Table demonstrate that the correlations for the separate subscales were highly similar to those obtained from the combined scales” (lines 195-200)

Below you will also find other minor details.

Supplementary material is not ordered in presentation order and sometimes is confusing.

Thank you for pointing this out. We have re-ordered the numbering of the supplementary materials to ensure they appear in a chronological order. For example we have renamed S5 Table to S2 Table as it appears earlier in the Results section. We have also made corresponding textual changes in the main manuscript to update the affected numbering.

Line 51. I think that in this case “QRP” should be QRPs.

Thank you. This change is made.

Line 68. RRP has been defined already in line 50.

Thank you. This change is made.

Line 70. In line 1, randomized response (RR) is mentioned but its importance is not explained. Later it is discussed again and here it is explained what it is. I would suggest eliminating RR here or anticipating its importance.

Thank you for pointing this out. We have included the explanation on why we chose to use the randomized response in the Introduction section in line 74-77 when it is first mentioned in order to address this comment. We removed it from line 382-3 to avoid repetition.

Lines 74-77 now read as follows: “It targets all academic researchers in The Netherlands across all disciplinary fields and uses a randomized response (RR) technique to assess engagement in FF as it is a well-validated method known to elicit more honest answers on highly sensitive topics [19].”

Line 89 to 93. I suggest adding a semicolon to separate the field of specialization from the position held.

This change is made.

Line 117. Why is it necessary to adapt de QRPs?

The QRPs originated from a survey done of participants from four World Conferences where nearly 60% of the surveyed participants came from the biomedical disciplinary field. As the NSRI targeted four disciplinary fields including those outside of the biomedical field, we wanted to ensure the QRPs from Bouter et al 2016 would remain applicable to our multidisciplinary target group. 

We have included lines 127-131 in the manuscript:

“The 11 QRPs were adapted from a recent study where 60% of the surveyed participants came from the biomedical disciplinary field [33]. As the NSRI targeted disciplinary fields including those outside of the biomedical field, we conducted a series disciplinary field specific focus groups to ensure the 11 QRPs from Bouter et al were applicable to our multidisciplinary target group. “

Line 134. Please define NA (like in 152). Not all readers are familiar with this nomenclature.

Thank you. This change is made.

Line 144. Why QRP has been dichotomized? Are there precedents of other authors doing it? If so, please give references.

Yes, there are precedents of this, see references 9 and 10 in the reference list (i.e. Xie Y et al 2021 and Fanelli D 2009). In order to make our results comparable to other studies, we therefore also dichotomized QRP. We have included these two references against this statement which now reads as follows:

“Prevalence was operationalized as the proportion of respondents who scored at least one QRP as 5, 6 or 7 to allow for comparability to other studies [9,10]” (line 177-79)

150. What multivariate analysis are you referring to?

We apologize for the lack of clarity. This refers to the multiple linear regression analysis shown in Tables 3a and 3b which we have now clarified as well in lines 175-177 which now reads as:

“In the multiple linear regression analysis (Tables 3a and 3b), overall mean QRP was computed as the average score on the 11 QRPs, after recoding not applicable scores to 1 (i.e. never)” 

154 Number 50 seems a reference. Maybe write “Fifty”

This change is made in lines 190 revised manuscript with track changes.

Line 182. I would add “see S1 table” for better reading.

This change is made to line 211, revised manuscript with track changes.

S1 table. Give more space in order to see full words in columns. Considering adding the N inside brackets: Life and medical sciences (N = 2746). It seems that in the cells there is the N (Column) and also de % (row).

This change is made to S1 Table.

S1a. In title add spaces in n = 8

This change is made to S1a.

S1b. In al flowchart add space after N = NUMBER.

This change is made to all flowcharts.

S1c and S2a to s2e. The image is pixelated. Maybe raising the dpi can make it better.

These images have been replaced with higher resolution ones.

S2b. Add sample size of subgroups. Probably this helps understanding the CI95% of undisclosed.

This change is made to S2b Table.

Table 1. “Chronbach’s alpha” is “Cronbach’s alpha”. Conisdering reducing the space between columns, and maybe add acronyms for better visualization.

Thank you for spotting the spelling error which is now corrected. We decided to not add acronyms as Reviewer 3 mentioned we had too many and suggested we reduce acronyms. To try to meet both reviewer comments, we decided to not add anymore new ones. 

Table 2. 

Please always use 1 decimal. For example, QRP1 in Arts and humanities 95%IC needs a 0.

This has now been thoroughly checked and corrected across all Tables including Table 2.

Reviewer #2: 

In the manuscript titled ‘Prevalence of questionable research practices, research misconduct and their potential explanatory factors: A survey among academic researchers in The Netherlands’ (PONE-D-21-28638), the authors conducted a study using the National Survey on Research Integrity to examine the prevalence of research misconduct, questionable research practices, and how these are associated with a variety of explanatory factors across various disciplinary fields and academic ranks in the Netherlands. The article is well-written, and the supplementary materials are helpful in further examining the survey and analyses. This article has potential to contribute to the existing literature on questionable research practices, however there are some issues that should be addressed.

Introduction

The introduction is well-written, and authors have provided a sound rationale for why this research is important. The authors also provided detailed explanations about the NSRI and its objectives in assessing QRPs, FF, and RRP.

Materials and Methods

The authors provided sufficient detailed information on the ethical review board approval. One advantage of this study is that the authors used the Open Science Framework to share their study protocol, ethics approvals, data analysis plan, and final dataset. This was helpful and important for reviewing this study, and for readers to gain a more in-depth understanding of the study questions and analysis plan.

Study Design

The authors provided clear inclusion criteria of eligible participants for the study and the ways eligible participants were notified about the study.

Survey Instrument

The authors presented clear explanations of the NSRI, the different components, and types of response options for the different sets of questions. It would be helpful to briefly summarize the types of questions for each of the components on page 6 lines 115-120, similar to the example provided for the explanatory factor scales.

Thank you for this comment which refers to the QRPs. We wish to clarify that the type of questions pertaining to the QRPs are mentioned in Table 2. Further to this, the open science framework (reference 30) contains the full verbatim questionnaire. For reasons of word limit and duplication, we therefore preferred to not mention the QRP questions in text in the manuscript again. However we have added an additional supplement S6 which now details all the questions in the NSRI and referenced in line 120-21 of the revised manuscript with track changes which now reads as 

“NSRI comprises of four components: 11 QRPs, 11 RRPs, two FFs and 12 explanatory factor scales (75 questions, detailed in S7)).”

Another advantage of this study was that authors used the randomized response technique to reduce social desirability and nonresponse. It could be helpful to briefly explain why the RR technique was only used with the FF questions (page 6 lines 118-120).

Thank you for this suggestion. The RR technique is best used for highly sensitive questions where it is shown to elicit more honest answers the higher the sensitivity of the content (see references 19 and 34) . As such we limited its use to the most sensitive questions on research misconduct i.e. falsification and fabrication. We have clarified this in lines 134-138 in the revised manuscript with track changes to read as follows:

“The RR technique is best known to elicit more honest answers, the more sensitive in nature the questions are [19,34]. Additionally, because the technique takes longer to apply, the survey would end up taking too long when all questions would use the technique. Hence, we chose to limit its use to only the most sensitive questions on research misconduct.”

Page 6 lines 122-126: Twelve explanatory variables including: scientific norms, peer norms, perceived work pressure, etc. were chosen. It would be beneficial to include reasons for why these twelve were chosen. Also, how many total explanatory variables are originally identified from the NSRI? It is not clear if the explanatory factors are comprised of 12 variables or if these were specifically chosen by the authors for a particular reason. 

The original selection of explanatory variables contained 13 scales. These were selected based on their use by experts in the field of research integrity and most cited in the literature. From these 13 scales, the authors of the NSRI made a further selection of the final 12 scales which were included in the NSRI based on the criterion that the scales included in the NSRI should address issues that would be actionable by research institutions. The scale “Organizational Justice of the field” was hence omitted for this reason. Lines 139-140 includes a sentence on this in the revised manuscript with track changes to read as follows:

“The explanatory factors scales were based on psychometrically tested scales most commonly used in the research integrity literature and focused on action-ability.”

Moreover, additional information would be important to include on the reason(s) why only the scientific norms description and normative behavior of peers were piloted (page 6, lines 128-129).

Thank you for your feedback. We wish to clarify that in addition to scientific norms subscription and peer norms, we also piloted the scales on competitiveness, likelihood of detection by collaborators, likelihood of detection by reviewers, and both organizational justice scales (procedural and distributional). This is indicated in lines 146-48. The remaining exploratory scales were used previously in highly similar samples (e.g., publication pressure; see ref 22) or in samples in earlier studies (ref 25,26) that we deemed to be sufficiently similar to our current sample. The Funding pressure scale was newly created and could not be piloted due to resource constraints. However, in the NSRI, this scale performed well in terms of psychometric properties (with a Cronbach alpha of 0.76 using six items) and in terms of convergent validity (i.e., positive correlations with publication pressure and competitiveness (S4 Table). 

We have included the following lines 148-154 in the manuscript to better explain these points:

“The other exploratory factor scales were either used previously in highly similar samples (e.g. publication pressure scale) [22] or in samples in earlier studies which were sufficiently similar to our current sample [25,26] except for the funding pressure scale which was newly created but could not be piloted due to resource constraints. However, in the NSRI, this scale performed well in terms of psychometric properties (with a Cronbach alpha of 0.76) and in terms of convergent validity (i.e., positive correlations with publication pressure and competitiveness (S4 Table).”

The authors conducted cognitive interviews to pre-test the NSRI comprehensibility, which can contribute to improving the survey. It would be interesting to note whether critical points were identified from the cognitive interviews that led to changes on the NSRI.

Thank you for this comment. Results on the cognitive interviews were centered around improving the clarity of certain words in the questionnaire, instructions, as well as feedback on the duration taken to answer the questionnaire. We have now added this information which reads as follows:

“In summary, the comments centered around improvement in layout such as the removal of an instruction video on the RR technique which was said to be redundant, improvement in the clarity of the instructions and to emphasize certain words in the questionnaire by use of different fonts for improved clarity. The full report of the cognitive interviews can be accessed at the Open Science Framework [30].” (lines 162-166)

Statistical analysis

Information on item analysis (e.g., whether items were deleted, modified, replaced due to low item-total correlation or item discrimination) are not provided. 

We have added the corrected item-total correlations of all items to S5 Table. In addition, no items had to be deleted as all items performed adequately in the NSRI.

In addition, Cronbach’s alpha values are provided to examine the reliability of the explanatory factor scale, but not with the other items/scales. This information would be beneficial to include across the different components of the NSRI or provide a rationale on why this was not included.

Thank you for raising this point. We wish to clarify that the other items being referred to are not psychometric scales. Any frequent QRP and Any FF are informative measurement scales that served as an index of the research behaviors at issue. Since they are not reflective measurements like the explanatory factor scales, we did not report Cronbach’s alpha values for them.

Minor comment: Page 7 lines 140-141 states that “Mean scores of individual QRPs only consider respondents that deemed the QRP at issue applicable.” It would be helpful to know the total number of participants who deemed the QRP an issue.

All univariate measures have been calculated over the applicable items only. The number of applicable cases per QRP have been added to the corresponding tables ( S2a and S2b Tables).

Results

Page 9 line 182 states: “There are about equal propositions of male and female respondents.” It would be beneficial to expand on this by including additional descriptives on the sample composition such as total number of participants across gender, disciplinary field, academic rank, and, if available, mean age, and average years spent in each academic rank.

Thank you for this clarification. We wish to point out that the additional descriptives asked for are included in S1 Table: Characteristics of all respondents by disciplinary field, academic rank, gender, research type and institutional support. Unfortunately, we did not collect information on age and average years spent in each academic rank for privacy protection reasons. We have now included a reference to S1 Table in line 223-224.

“There are about equal proportions of male and female respondents. Further breakdown by disciplinary field, academic rank, research type and institutional support is detailed in S1 Table.”

Page 9 lines 188-190 states: “Postdocs and assistant professors report the highest scale score for publication pressure (4.2), funding pressure (5.2), and competitiveness (3.7), and the lowest scale score for organizational justice (4.1) (Table 1).” However, based on Table 1, it appears that the highest scale scores for postdocs and assistant professors are: scientific norms (6.2), funding pressure (5.2), and work pressure (4.7), and the lowest scale score for this group was likelihood of detection (collaborators) (3.5). 

We wish to clarify that the scale scores for postdocs and assistant professors are made in comparison to the other two academic ranks. We apologize that this was not clear in the original text. We have now amended this and also included peer norms which we inadvertently left out in addition to organizational justice as the lowest scales’ scores for this rank.

We have clarified and correct this in line 230-233of the revised manuscript with track changes which now reads as:

“Postdocs and assistant professors report the highest scale scores for publication pressure (4.2), funding pressure (5.2) and competitiveness (3.7), and the lowest scale score for peer norms (4.1) and organizational justice (4.1) when compared to the other academic ranks (Table 1).”

In addition, it also appears that the highest scale scores for the respondents from the arts and humanities includes scientific norms (6.2), work pressure (4.8), and funding pressure (4.6), with the lowest including mentoring (3.5) and likelihood of detection (collaborators) (3.5).

Similarly, the scale scores for the arts and humanities are again in comparison to the three other disciplinary fields and are therefore correct except for the omission of peer norms which should have been included as one of the lowest scale scores. We apologize for any confusion on this. We have clarified and correct this in lines 233-237 of the revised manuscript with track changes:

“Respondents from the arts and humanities have the highest scale scores for work pressure (4.8), publication pressure (4.1) and competitiveness (3.8). They also have the lowest scores for mentoring, peer norms and organizational justice (3.5, 4.1 and 3.9, respectively) when compared to the other disciplinary fields (Table 1).”

Minor comment: It would be beneficial to include a description on how junior researcher was operationalized for the purposes of the study (e.g., two years post PhD).

In the Netherlands, junior researchers are defined as individuals with a Masters or PhD degree and doing a minimum of 8 hours per week research tasks under close supervision of a senior researcher (i.e. postdoc, assistant, associate or full professor). We have included this description in lines 100-102 which now reads as follows:

“To be eligible, researchers had, on average, to do at least 8 hours of research-related activities weekly, and belong to life and medical sciences, social and behavioural sciences, natural and engineering sciences, or the arts and humanities; and be a PhD candidate or junior researcher (who is defined in The Netherlands as an individual with a Masters or PhD degree doing a minimum of 8 hours per week of research related tasks under close supervision), postdoctoral researcher or assistant professor or associate or full professor.

Discussion

If results from page 9 lines 188-190 are revised, page 14 lines 280-281 should also be revised to reflect these changes (e.g., postdocs and assistant professors report highest scale score to be scientific norms, funding pressure, and work pressure).

Thank you for following up on this comment. We have checked this and wish to mention that the highest and lowest scale scores for postdocs and assistant professors is correct as it is a comparison between ranks. Therefore this is correct except for the inclusion of peer norms as one of the lowest scoring scales which we inadvertently omitted. The affected lines 324-328 now read as follows

“Postdocs and assistant professors rate publication pressure, funding pressure and competitiveness higher than other academic ranks, but peer norms and organizational justice lower. Arts and humanities scholars reported experiencing the highest work and publication pressures, the most competition and the lowest in mentoring, peer norms and organizational justice compared to other disciplinary fields.”

Strengths and Limitations

Authors noted critical limitations to the study (e.g., coding NA responses as ‘never’ for statistical analyses and definition of “any frequent QRP” response associated with scores of 5, 6, or 7). These are important limitations that could potentially affect the results of the study.

Reviewer #3: 

Dear Authors,

Thank you for your interest in PLOS ONE. I very much enjoyed reading the manuscript as it emphasizes on the areas for improvement in research practices in academe. I have some suggestions and feedback for your consideration.

General comments:

• You have many abbreviations, which, I think, makes it hard to read. I would suggest no more than 2-3 abbreviations throughout the manuscript to help your reader follow your ideas.

Thank you for your suggestion and feedback. We have carefully reviewed this suggestion and balanced it against that of reviewer 1 who requested to include more abbreviations. We have removed three abbreviations in the manuscript and provided them in full instead when relevant. These are: WMO, OSF and UMC. We hope to have adequately accommodated both reviewers’ requests.

• Line 154, you start a sentence with a number. Please consider writing it out.

Thank you. This change is made in line 190 of the revised manuscript with track changes.

Introduction:

• Line 47: The very first sentence is extremely broad and does not give us enough information about the kind of “important challenges.” Please explicate that part after challenges to prepare your reader about the topic.

Thank you for the feedback. We have revised this text in lines 47-50 of the revised manuscript with track changes to read as follows:

 “The basis of sound public policy relies on trustworthy and high quality research [1]. This trust is earned by being transparent and by performing research that is relevant, replicable, ethically sound and of rigorous methodological quality.”

• Line 50: There is a transition sentence needed after the sentence ends with “…methodological principals” and before “To promote responsible research practices…” to make a connection between two different ideas. In addition (Line 50), you start the sentence with “responsible research practices” and “questionable research practices” without any explanation about its meaning. I would suggest explaining these concepts before you move on to the problem and gap.

Thank you for this suggestion. We have now revised the affected lines such that they read as follows in lines 53-56 of the revised manuscript with track changes:

“Continued efforts to promote responsible research practices (RRPs) which include open science practices like open data sharing, pre-registration of study protocols, open access publication over questionable research practices (QRPs) are therefore needed. In order to support the need for such continued efforts, solid evidence on the prevalence of research misconduct and QRPs as well as the factors promoting or curtailing such behaviours are needed.”

Study Design:

• I had hard time to understand your classification of participants as PhD candidate or junior researcher, postdoctoral researcher or assistant professor, or associate or full professor. A justification (along with a reference) is needed to explain how and why you considered this categorization and what you mean by “junior researcher.” Do you mean researchers, such as data analysts, who have degrees at master’s level OR you paid attention to their years of experience as researcher? Clarification is needed.

Thank you for raising this point. While we do not have reference for this, the definition in academia in the Netherlands for junior researchers is individuals with a Masters or PhD degree and doing a minimum of 8 hours per week research tasks under close supervision of a senior researcher (i.e. postdoc, assistant, associate or full professor). We have included this description in lines 97-102 in the revised manuscript which now reads as follows:

“To be eligible, researchers had, on average, to do at least 8 hours of research-related activities weekly, and belong to life and medical sciences, social and behavioural sciences, natural and engineering sciences, or the arts and humanities; and be a PhD candidate or junior researcher (who is defined in The Netherlands as an individual with a Masters or PhD degree doing a minimum of 8 hours per week of research related tasks under close supervision), postdoctoral researcher or assistant professor or associate or full professor.”

• Regarding your classification of participants based on their status (e.g., PhD candidate, postdoc, etc.), your survey involves items that asked about their actions over the previous three years, which needs justification (please consider adding a brief sentence for why the last three years not five, for example). So, when you report your findings, you actually report on their previous positions. For example, whatever a postdoc provided answers about in the survey actually refers to his/her activities as a PhD candidate unless the postdoc has the postdoc position for more than 3 years. Therefore, not only what their status (e.g., assistant professor, post doc) is but also the years they have been in that position matters and affects the interpretation of findings. Otherwise, whole findings are misleading. If you have their years of experience in that position, please consider adding that variable in the analysis to account for retrospective effect and report accordingly. If you do not have that variable in the survey, meaning no data collected about their years in their current position, then I would suggest acknowledging this as limitation and adding a note when interpreting findings.

Thank you for raising these points. We chose to ask respondents about their behaviours in the last three years as this is an acceptable timeframe to limit recall bias that is also used in a number of other surveys on this subject (see references 9 and 10). We have included a sentence to this effect in lines 126-27 of the revised manuscript which now reads as

“All respondents obtained the same set of questions on QRPs, RRPs and FF, referring to one’s behavior in the previous three years. A three year timeframe was chosen to limit recall bias and is also a timeframe used in other similar studies [9,10].”

On the issue of misclassifications: some misclassifications may occur as you correctly point out. However as we did not collect this type of data due to the strict privacy design of the survey especially in disciplines with smaller sizes such as the humanities, we are unable to determine the extent of its effect on our results. We expect it to be minor though given promotion rates in our experience and in our networks in academia. We have therefore included a few sentences on this in the Limitations section which reads as follows:

“Another potential limitation we wish to mention is misclassifications in academic rank due to promotion of individuals to a higher rank less than 3 years prior to completing our survey. Their responses are therefore likely to partly represent their behaviors whilst in a lower academic rank. However we did not collect information on years of experience of respondents in a rank due to strict privacy design of the survey. As such we are unable to comment on the impact of this misclassification on our results but we believe it to be relatively minor. Future surveys on this topic may, however, wish to take this into account in their design and analysis.” (lines 441-447).

Survey Instrument:

• Line 119, you note that randomized response technique was used but you do not explain what it is until later in the manuscript (Lines 375-376). I would suggest moving that explanation to the survey instrument section to help your reader understand your justification and what this technique is/does.

Thank you for this suggestion. We have included the explanation on why we chose to use the randomized response in the Introduction. We removed it from line 375 to avoid repetition.

“It targets all academic researchers in The Netherlands across all disciplinary fields and uses a randomized response (RR) technique to assess engagement in FF as it is a well-validated method known to elicit more honest answers on highly sensitive topics [19].” (revised manuscript with track changes lines 74-77)

• Lines 135-137, you provide a statement about the pre-testing the NSRI questionnaire through cognitive interviews. This should be moved up on that page before you explain the number of total items and other details (starting after the sentence Line 129 or adding a paragraph before Line 130). I would suggest adding a short paragraph and explaining whether you made any changes on the questionnaire based on cognitive interviews. If that information was captured in your previous study, then consider citing it with a brief explanation.

Thank you for this suggestion. Results on the cognitive interviews were centered around improving the clarity of certain words in the questionnaire, including instructions, as well as feedback on the duration taken to answer the questionnaire. We have now added this information in lines 162-66 of the revised manuscript with track changes which now read as:

“In summary, the comments centered around improvement in layout such as the removal of an instruction video on the RR technique which was said to be redundant, improvement in the clarity of the instructions and to emphasize certain words in the questionnaire by use of different fonts for improved clarity. The full report of the cognitive interview can be accessed at the Open Science Framework [30].”

Statistical Analysis:

• I had hard time to understand why you chose to compute average score on the 11 QRPs after recoding not applicable scores to 1. That skews your findings and does not accurately reflect participants’ attitudes because it was not applicable to them. I would suggest re-running the analysis after coding not applicable as zero (0). It should not be included in the sum or average score. You can explain this with a brief sentence and note that range of total score in the survey.

Thank you for this comment. We wish to clarify that the “not applicable” values are bonafide missing values. While we understand that recoding the them to zero may seem semantically intuitive, there are valid statistical and procedural reasons why we chose to replace these values with the lowest observed category (1 = Never). Please allow us to explain these here: 

First, the recoding of “not applicable” to 1 is part of our pre-registered data analysis plan. Second, we did run extensive sensitivity analyses to study the validity of our pre-registered choice. Based on these analyses we concluded that our pre-registered choice is the most valid solution to this issue in this data set. These sensitivity analyses can be found in the OSF data analysis folder> subfolder entitled “Figures and Tables> Table 3 Regressions”: https://osf.io/ehx7q/.

Third, replacing “not applicable” with the value 0 is not a bonafide value that could have been observed. Using non-bonafide constants to fill in missing values is unreliable and statistically invalid. As such filling in zero would underestimate any parameters to a much greater extent than using a bonafide observed value would. We believe inducing such deliberate bias would therefore be undesirable. Fourth, coding “not applicables” as zero yields a positive correlation between QRP and RRP in a confirmatory factor analysis. This is counterintuitive and not in line with theoretical, nor practical expectations. Coding the “not applicables” as 1 (or any other bonafide observed value, for that matter) yields an expected negative correlation between the factors QRP and RRP. 

Lastly, the validity of our pre-registered data analysis plan with respect to the “not applicable” has been confirmed by two independent replications on two different data structures reference: De Koning and Van der Sluis (2021). Modeling not applicable answers when data are incomplete. Master Thesis. Utrecht University.

To ensure this is also explained in the Limitations section of the manuscript, we have included this change in lines 432-435:

“We have studied other recodes of the NA answers and remain confident that our preregistered choice yields inferences that do not ignore the non-random distributions of the NA answers and do not violate theoretical and practical expectations about the relation between QRP and other studied practices.”

Results:

• Line 176: You mention eight universities supported the NSRI but do not provide information about non-supporting universities until later in the manuscript. An explanation about why they did not support, if known, would be helpful to contextually understand the findings as there might be some bias or buy-in to the work you did, which affects your findings as this cannot be generalizable in the context of the Netherlands.

The reasons why the non-supporting institutions refused to join the NSRI are not clear to us either. Therefore we are unable to provide a sound rationale for this non participation of some but not others. Some probable reasons for this non -participation are alluded to in this Science Magazine article about our study: https://www.science.org/content/article/largest-ever-research-integrity-survey-flounders-universities-refuse-cooperate. In our experience in the design and conduct of this study over the last three years, and expertise in research integrity generally, the topic of research integrity remains highly sensitive with a fear for bad reputation/press among research institutions at large.

As to any bias which may affect the study’s generalizability, we cannot exclude this as you correctly point out which we also addressed in our Limitations section lines 420-27. But given that our main findings align well with other international research integrity surveys we believe our results are generalizable and any bias, if present, may be more an underestimate based on the premise that a survey like this may tend to attract respondents with an interest to improve research integrity. 

• A brief description of the word “prevalence” would be helpful in the context of the study to help reader understand what you mean. 

Line 203, you note “The five most prevalent QRPs…” where you mean highly scoring items overall, whereas Line 57, you note about “QRPs becoming alarmingly prevalent” where you emphasize the commonness of the problem. These are two different things and need to be clarified.

Thank you for the suggestion. We defined prevalence in this study as the percentage of respondents who chose 5,6, or 7 (i.e. sometimes to frequently) on the Likert scale for a QRP divided by the total number of respondents who responded to that QRP. We have included a definition of this in lines 177-79:

“Prevalence was operationalized as the proportion of respondents who scored at least one QRP as 5, 6 or 7 over the total respondents for that QRP. This definition will allow for comparability to other studies [9,10].”

Discussion:

• The first sentence (Lines 274-275) needs citations of others’ work.

We have added the citations to line 319 of the revised track changes manuscript:

“Our research integrity survey among academics across all disciplinary fields and ranks is one of the largest worldwide [9,10].”

• Line 310, “….journal reviewers or otherwise” what do you mean by “otherwise?”

Thank you for this question. This refers to other types of peer review outside of journal peer review such as post publication peer review or scholarly reviews on preprints and similar. We have added this clarification to lines 353-356 revised track change manuscript:

“The increased transparency offered by open science practices such as data sharing, is likely to boost chances of detection of research misconduct whether through formal journal reviewers or otherwise such as through post publication peer review or other types of scholarly reviews such as comments on preprints [25].”

• Lines 314-317, add a sentence after this sentence to explain the connection between the earlier study mentioned and your study.

Thank you for this suggestion. We have made changes to the sentences such that they now link better. They read now as:

“Yet, surprisingly, we find a moderate yet statistically significant association between mentoring and higher overall QRP mean suggesting that mentoring may be associated with higher QRPs. An earlier study [13] explored five different types of mentoring (including responsible and survival mentoring that we measured) and suggested that mentors can influence behaviour in ways that both increase and decrease the likelihood of problematic behaviours which may explain the negative association we see with mentoring in our study.” Lines 361-367.

• The concept of “sloppy science” is used with and without quotation mark. Is there a reference for this concept in using quotation mark?

Thank you for this question. It is a term used interchangeably with questionable research practices (QRP) in the field of research integrity studies. For consistency, we have replaced this with QRPs which is now the term we use throughout the manuscript.

• Line 356: the statement needs citation as you reference to a smaller study.

The reference was included in the line below. We have moved it to the line above i.e. line 407, revised track changes manuscript.

Strengths and limitations:

• Lines 365-366, you note that you were unable to credibly verify eligibility criteria based on emails. Didn’t you ask eligibility questions? I could not understand why this would be an issue if you asked eligibility questions.

Thank you for this clarification. We are referring to not being able to verify the eligibility of the email addresses of the non-supporting institutions against our survey eligibility criteria because we had to web scrape the email address for the non-supporting institutions. Thus we could not assess eligibility of addresses scraped a priori. This is what we make reference to in these lines. We could only determine eligibility from these institutions when a respondent opened the survey link and filled out the eligibility questions. As a consequence, we could only reliably calculate the response for the supporting institutions. This is the point we are explaining in these lines. We hope this explanation sufficiently addresses the confusion. We have revised lines 415-419 as well to read as follows:

“The email addresses of researchers affiliated to non-NSRI-supporting institutions were web-scraped from open sources. Therefore, we are unable to credibly verify if the scraped email addresses matched our eligibility criteria prior to participation in the survey. Hence, we could only reliably calculate the response to the NSRI based on the eight supporting institutions”

• Lines 366-367, you note that you calculated responses only based on eight supporting institutions. Then, what is the point of reporting non-supporting institutions on Lines 185-188?

Lines 185-188 in the original manuscript make reference to the distribution of respondents over the four disciplinary fields. We included this information as descriptive information on how evenly distributed (or not) the four disciplinary fields were in our respondent population.

Lines 366-367 of the original manuscript refers to how we calculated the response of the survey.

We hope this sufficiently clarifies your query.

---

## [Decision Letter · Decision Letter 1]

1 Dec 2021

PONE-D-21-28638R1Prevalence of questionable research practices, research misconduct and their potential explanatory factors: a survey among academic researchers in The NetherlandsPLOS ONE

Dear Dr. Gopalakrishna,

Thank you for submitting your manuscript to PLOS ONE. After careful consideration, we feel that it has merit but does not fully meet PLOS ONE’s publication criteria as it currently stands. Therefore, we invite you to submit a revised version of the manuscript that addresses the points raised during the review process.

We look forward to receiving your revised manuscript.

Kind regards,

Sergi Fàbregues

Academic Editor

PLOS ONE

Journal Requirements:

Reviewers' comments:

Reviewer's Responses to Questions

**Comments to the Author**

1. If the authors have adequately addressed your comments raised in a previous round of review and you feel that this manuscript is now acceptable for publication, you may indicate that here to bypass the “Comments to the Author” section, enter your conflict of interest statement in the “Confidential to Editor” section, and submit your "Accept" recommendation.

Reviewer #1: (No Response)

Reviewer #2: All comments have been addressed

Reviewer #3: All comments have been addressed

2. Is the manuscript technically sound, and do the data support the conclusions?

Reviewer #1: Yes

Reviewer #2: Yes

Reviewer #3: Yes

3. Has the statistical analysis been performed appropriately and rigorously? 

Reviewer #1: Yes

Reviewer #2: Yes

Reviewer #3: Yes

4. Have the authors made all data underlying the findings in their manuscript fully available?

Reviewer #1: Yes

Reviewer #2: Yes

Reviewer #3: Yes

5. Is the manuscript presented in an intelligible fashion and written in standard English?

Reviewer #1: Yes

Reviewer #2: Yes

Reviewer #3: Yes

6. Review Comments to the Author

Reviewer #1: As I commented in the first review, I find this article very attractive and it has a lot of potential to be published. In this case, besides some minor changes, I think that one of the suggestions made in the first review should be reconsidered.

(1) The authors consider that they have added the psychometric properties since they have added in the supplementary materials (as also suggested by another reviewer) the correlation of the items with the total. I believe that this is insufficient considering the large sample size. I suggest that the authors make a final analysis by performing a factor analysis (if possible confirmatory) in order to justify the use of scale scores.

(2) In addition, along the same lines, the decision to combine the two scales (Mentoring and Organizational Justice scales) should also be justified with factor analysis techniques. Despite this comment, I believe that the text added by the authors helps to understand what happened. We cannot make a decision about the dimensionality of the scale just based on the value of the correlation.

Below are some other details in case you would like to review them.

(1) NSRI comprises of four components: 11 QRPs, 11 RRPs, two FFs and 12 explanatory factor 118 scales (75 questions, detailed in S7). S7 does not exist. Revise number.

(2) Check brackets in line 154: they do not close.

(3)It is confusing to me that there is a Figure S2, and S2a-8 are not related to those figures.

(4) Figure 1: a)A space is missing between N and in equal in all rectangles. B)It would be necessary that the lines start touching the rectangle and not floating.

(5) Table 1. When result is 0, please indicate 0.00

(6) Table 3: I do not understand the symbols in the title as I cannot find it inside the table. If it is already in the foot, it is not necessary to add it in the title.

Reviewer #2: Thank you for considering my suggestions and incorporating them in the revised manuscript (PONE-D-21-28638-_R1). The authors have adequately addressed all my comments and provided sufficient information. There are only a few minor clarifying/formatting comments.

I wish you much success on your future work. Thank you.

Page 6 line 121: This sentence references S7, but I am not sure it is included in this file or if it is only included on the Open Science Framework. I might have missed this, but I just want to double-check.

Page 7 line 153: States “Cronbach alpha,’ instead this should read “Cronbach’s alpha.”

Page 7 line 154: If using APA 7th edition, APA suggests using brackets instead of double parentheses: “(i.e., positive correlations with publication pressure and competitiveness [S4 Table]).”

Page 9 line 197: Similar to above: “(correlation factor of > 0.8 [S4 Table]).”

Reviewer #3: Dear Authors,

I appreciate your responsiveness to the feedback. Thank you for addressing the comments.

7. PLOS authors have the option to publish the peer review history of their article (what does this mean?). If published, this will include your full peer review and any attached files.

Reviewer #1: No

Reviewer #2: No

Reviewer #3: No

---

## [Author Response · Author response to Decision Letter 1]

22 Dec 2021

Reviewer #1: As I commented in the first review, I find this article very attractive and it has a lot of potential to be published. In this case, besides some minor changes, I think that one of the suggestions made in the first review should be reconsidered.

(1) The authors consider that they have added the psychometric properties since they have added in the supplementary materials (as also suggested by another reviewer) the correlation of the items with the total. I believe that this is insufficient considering the large sample size. I suggest that the authors make a final analysis by performing a factor analysis (if possible confirmatory) in order to justify the use of scale scores.

(2) In addition, along the same lines, the decision to combine the two scales (Mentoring and Organizational Justice scales) should also be justified with factor analysis techniques. Despite this comment, I believe that the text added by the authors helps to understand what happened. We cannot make a decision about the dimensionality of the scale just based on the value of the correlation.

We wish to thank the reviewer for making this excellent point on the dimensionality of the scales. Accordingly, we ran a confirmatory one-factor model in Lavaan using the Weighted Least Squares Mean and Variance (WLSMV) estimator. We report these results for the explanatory factor scales in S5 Table Supplementary information. 

As can be seen, the one-factor analysis showed excellent fit for all but one of the scales, with only the fit of the Competitiveness scale highlighting some misfit (which on inspection was caused by three items clustering together). As this is an unexpected finding, and not previously part of our preregistered data analysis plan, we therefore will consider doing explorative analysis with this finding at a later stage.

We also ran two and one factor models for the mentoring and organizational justice scales that we previously merged to get a sense of the dimensionality, as rightly suggested by the reviewer. In the analyses of the two original mentoring scales, we found the two factor model to fit appreciably better (CFI: 0.99, RMSEA: 0.062, SRMR: 0.051, correlation between factors>.70) than the one-factor model (CFI: 0.97, RMSEA: 0.105, SRMR: 0.086). Hence, we decided to re-run all analyses in the main manuscript that treated Survival Mentoring and Responsible Mentoring as separate explanatory factor scales. 

As can be seen in the results of the linear prediction of QRP mean in Table 3b, this led to more interpretable results in which -as expected and inline with pertaining literature- Survival Mentoring was positively related to QRP mean, while Responsible Mentoring was negatively (albeit non-significantly) related to mean QRP use. We also fitted two and one factor models to the two original organizational justice scales (results of the separate factor analyses in S5 Table already highlighted that each 6-item scale separately fitted a one-factor model). However, in the full model with all 12 organizational justice items, the two factor model that separated the distributional justice and procedural justice (CFI: 0.99, RMSEA: 0.046, SRMR: 0.043, correlation between factors: >.98) fitted just as well as the one-factor model (CFI: 0.99, RMSEA: 0.046, SRMR: 0.043). In light of the very high correlation between the two factors and to avoid multi-collinearity, we decided to focus on the joint 12-item organizational justice scale in the main analyses (as we did in the original submission).

In accordance to these results based on the analyses above, we have adjusted the following parts of the manuscript:

• Text in the Methods section, lines 188-191 where we removed reference to merging the mentoring scale, and in the Discussion lines 352-60 which read as follows respectively, 

Methods (lines 188-91): “The subscales distributional and procedural organizational justice were highly correlated (correlation factor of >0.8 [S4 Table]).”

Discussion (lines 352-60): “Unsurprisingly, we find a moderate yet statistically significant association between survival mentoring and higher overall QRP mean suggesting that survival mentoring may be associated with higher QRPs while an association in the opposite direction, again moderate but significant, is observed for responsible mentoring and overall QRP mean. Both results agree with an earlier study [13] which explored five different types of mentoring (including responsible and survival mentoring). Our study and that of Anderson et al [13] suggests that mentors can influence behaviour in ways that both increase (in the case of survival mentoring) or decrease (in the case of responsible mentoring) the likelihood of problematic research behaviour such as QRPs

• Tables 1, 3a and 3b (Main manuscript) have also been modified based on the two mentoring subscales analyzed now as two separate scales in the regression analyses.

• Supplementary information: S5 Table is also now revised to include the two mentoring subscales and the item correlations and factor loadings for all scales respectively.

Below are some other details in case you would like to review them.

(1) NSRI comprises of four components: 11 QRPs, 11 RRPs, two FFs and 12 explanatory factor 118 scales (75 questions, detailed in S7). S7 does not exist. Revise number.

Thank you for spotting this mistake. We have changed the numbering as this should refer to S6.

Lines 117 -118 now read as:

“NSRI comprises of four components: 11 QRPs, 11 RRPs, two FFs and 12 explanatory factor scales (75 questions, detailed in S6)”

(2) Check brackets in line 154: they do not close.

We have now corrected this at Line 154.

(3)It is confusing to me that there is a Figure S2, and S2a-8 are not related to those figures.

We have now changed Figure S2 to Figure S3 so it is separate in numbering from S2a-2e Figures.

(4) Figure 1: a)A space is missing between N and in equal in all rectangles. B)It would be necessary that the lines start touching the rectangle and not floating.

Thank you. These changes have been made to the affected Figure 1.

(5) Table 1. When result is 0, please indicate 0.00

These changes have been made to Table 1.

(6) Table 3: I do not understand the symbols in the title as I cannot find it inside the table. If it is already in the foot, it is not necessary to add it in the title.

Thank you for correctly pointing this out. All symbols are now only indicated in the table and their explanations as footnotes.

Reviewer #2: Thank you for considering my suggestions and incorporating them in the revised manuscript (PONE-D-21-28638-_R1). The authors have adequately addressed all my comments and provided sufficient information. There are only a few minor clarifying/formatting comments.

I wish you much success on your future work. Thank you.

Page 6 line 121: This sentence references S7, but I am not sure it is included in this file or if it is only included on the Open Science Framework. I might have missed this, but I just want to double-check.

Thank you for this comment. S7 should have been S6. This was a numbering mistake on our part which we have now corrected in the manuscript lines 117-118.

Page 7 line 153: States “Cronbach alpha,’ instead this should read “Cronbach’s alpha.”

This change has been made to line 151 of the track change manuscript.

Page 7 line 154: If using APA 7th edition, APA suggests using brackets instead of double parentheses: “(i.e., positive correlations with publication pressure and competitiveness [S4 Table]).”

Thank you. We have removed the double parentheses and replaced with brackets as suggested above. 

Page 9 line 197: Similar to above: “(correlation factor of > 0.8 [S4 Table]).”

Thank you. We have removed the double parentheses and replaced with brackets as suggested above. 

Reviewer #3: Dear Authors,

I appreciate your responsiveness to the feedback. Thank you for addressing the comments.

---

## [Decision Letter · Decision Letter 2]

11 Jan 2022

Prevalence of questionable research practices, research misconduct and their potential explanatory factors: a survey among academic researchers in The Netherlands

PONE-D-21-28638R2

Dear Dr. Gopalakrishna,

We are pleased to inform you that your manuscript has been judged scientifically suitable for publication and will be formally accepted for publication once it meets all outstanding technical requirements.

Kind regards,

Sergi Fàbregues

Academic Editor

PLOS ONE

**Comments to the Author**

1. If the authors have adequately addressed your comments raised in a previous round of review and you feel that this manuscript is now acceptable for publication, you may indicate that here to bypass the “Comments to the Author” section, enter your conflict of interest statement in the “Confidential to Editor” section, and submit your "Accept" recommendation.

Reviewer #1: All comments have been addressed

2. Is the manuscript technically sound, and do the data support the conclusions?

Reviewer #1: Yes

3. Has the statistical analysis been performed appropriately and rigorously? 

Reviewer #1: Yes

4. Have the authors made all data underlying the findings in their manuscript fully available?

Reviewer #1: Yes

5. Is the manuscript presented in an intelligible fashion and written in standard English?

Reviewer #1: Yes

6. Review Comments to the Author

Reviewer #1: First of all, I would like to thank the authors for their efforts to incorporate the suggestions made. I leave some very minor comments to be considered by the authors and/or for the person editing the manuscript if it is accepted.

Line 171. “In the” -> in the

Line 204/384. i.e., should be ¨that is¨

Line 212 -> I do not understand what is “ the response” here. Maybe is missing an adjective or a verb? Response rate?

Line 234: the character “;” is repeated twice.

7. PLOS authors have the option to publish the peer review history of their article (what does this mean?). If published, this will include your full peer review and any attached files.

Reviewer #1: No

---

## [Editor Report · Acceptance letter]

24 Jan 2022

PONE-D-21-28638R2 

Prevalence of questionable research practices, research misconduct and their potential explanatory factors: a survey among academic researchers in The Netherlands 

Dear Dr. Gopalakrishna:

I'm pleased to inform you that your manuscript has been deemed suitable for publication in PLOS ONE. Congratulations! Your manuscript is now with our production department. 

Kind regards, 

on behalf of

Dr. Sergi Fàbregues 

Academic Editor

PLOS ONE